# Research on parameterized modeling and mechanical characteristics of shearer cables

**Lijuan Zhao[1,2,3], Haining Zhang[1]\*, Feng Gao[4], Shijie Yang[1]**

1 School of Mechanical Engineering, Liaoning Technical University, Fuxin, China, 2 The State Key Lab of Mining Machinery Engineering of Coal Industry, Liaoning Technical University, Fuxin, China, 3 Liaoning Province Large Scale Industrial and Mining Equipment Key Laboratory, Fuxin, China, 4 Shandong Yankuang Group Changlong Cable Manufacturing Co., Ltd, Jining, China

\* jscqtzhn@126.com

**Data Availability Statement:** Tensile test results for conductor strands and insulation and sheath rubber of shearer cables are available from the kaggle database. (URL:https://www.kaggle.com/

## Abstract

Shearer cables, subjected to large deformations and exposed to harsh working conditions during frequent back-and-forth movements, pose difficulties in achieving comprehensive mechanical performance and extended fatigue life. This study addresses parametric modeling challenges related to determining tangency within and between layers, recursively generating spiral curves from the (n-1)-th level to the n-th level, and constructing irregular surfaces for insulation and sheath. Investigating tensile and bending properties, the research explores the impact of varying pitch diameter ratios at different stranding levels, stranding directions, and monofilaments on mechanical performance across scales. The results reveal a nonlinear increase in stress in power and control conductors with growing pitch diameter ratios. The optimal combination is determined as a pitch diameter ratio of 6 for cabling, 5 for the control conductor, and 14 for the power conductor. The predicted fatigue life of the improved cable by Ncode aligns with bending test results, demonstrating functionality up to $15.12e^4$ cycles. Stress distribution in parallel stranding is lower and more even, tending to scatter. Conversely, counter stranding experiences relatively higher stress, ensuring a more stable structure. While maintaining a constant effective conductor cross-sectional area, finer monofilaments result in higher cross-sectional filling ratios, enhancing tensile and bending performance.

## 1. Introduction

The shearer cable, a mobile rubber-sheathed flexible cable designed for coal mining, forming a multi-layered and spiral-stranded structure from diverse materials. Modeling challenges involve determining tangency within and between layers of the stranded structure, deriving equations for the n-level spiral curves, and constructing the insulation and sheath with surfaces of varying morphology. Consequently, there is a pressing need to investigate advanced modeling approaches for this intricate cable structure. Qian et al. [1] established a three-dimensional geometric model of 1860 grade 7-wire strand in Solidworks based on the parametric equations of cylindrical spiral curves in Cartesian space. Wu et al. [2] derived the spatial vector expression of a secondary helix in the Frenet-Serret coordinate system, determining the helical

datasets/ninghaizhang/tensile-test-of-shearer-cables).

**Funding:** This work was supported by the National Natural Science Foundation of China [Grant number 51674134]. The funders had no role in study design, data collection and analysis, decision to publish, or preparation of the manuscript.

**Competing interests:** The authors have declared that no competing interests exist.

direction based on the sum and difference of rotation angles. A 6×36WS-IWRC type wire rope model was established in Creo using the curve equation and helical sweep command. Erdön-mez [3], based on the parameter equations for single and double helix structures, derived expressions for the centerline, curvature, and torsion. These expressions were extended to triple helix and higher n-order helix structures. Jia and Du [4, 5] introduced a bottom-up approach to develop a three-dimensional model for a multi-level superconducting cable using discrete elements. By implementing effective contact force laws among distinct functional material units, the authors explored localized deformations and non-uniform contact mechanics within internal sub-cables, extending their analysis down to the scale of individual strands. Their investigation encompassed intricate external forces and diverse operational conditions. Study on modeling for shearer cables, Li et al. [6] used Solidworks to construct the power-frequency shearer cables both for the self-powered system and the coal mine gallery. The cable structure was simplified without copper shielding layers, and the simplified motor model was simulated in the magnetic field environment in Comsol software. Zhao et al. [7] took the strand as the smallest analytical unit, established the cylindrical coordinate equation for the first-level helical centerline in cabling, utilized the trajpar function to dimensionally drive the cross-section, and generated a second-level helical stranded structure perpendicular to the first-level helical centerline. Bai [8] derived equations for the centerline coordinates (x, y, z) of the power conductor with respect to the parameter t. The formula for the scanning section size was expressed by the pitch, and changing the phase difference generated the centerlines of the other two power conductors. Based on this, the parameter expressions of the centerlines of the four control conductor cores with higher stranding levels were obtained. The research methods mentioned above have certain limitations in the parametric modeling of stranded structures. As the stranding level (n≥3) continues to increase, the equations for the n-level spiral become exceptionally complex, posing challenges for both solving the equations and further development. Moreover, the above modeling methods do not account for the changes in the normal plane direction during the stranding process. Additionally, there is a lack of investigation into the construction of heteromorphic surfaces for insulation and sheath.

The mechanical properties of stranded structures are mainly associated with their tensile and bending performance. Studying the mechanical behavior of wire ropes, submarine cables, and superconducting cables holds reference significance for researching the mechanical characteristics of the shearer cable in this paper. Taking the wire rope structure as the research object, Liu et al. [9] investigated the impact of the twist pitch on the tensile mechanical properties of the first-level helical structure in anchor cables and its anchoring effect. They established the relationship between the twist pitch and its ultimate tensile load, as well as the maximum anchoring force, and identified the optimal twist pitch. Xiang et al. [10] investigated the mechanical behavior of multi-strand wire ropes with different lay directions under axial loads. Shao [11] established a model for the bending state of a wire rope, applied a force load tangent to the sheave groove to achieve the target bending radius, and analyzed the factors influencing the fatigue failure of the wire rope under bending. Focusing on submarine cables, Li, Wang, Lü, et al. [12–14] conducted tensile tests and finite element simulations on armored optical fiber composite submarine cables. They analyzed the relative movement and stress distribution patterns of each layer during cable laying and twisting under external forces. Dong, Zhang, Li, et al. [15–17], dedicated to CICC superconducting cables, utilized a differential geometry analytical approach to compute the equivalent modulus of strands. By integrating tensile tests with finite element numerical simulations and considering local contact deformations and friction between strands, they explored the impact of temperature, pitch, and friction on the mechanical characteristics of both primary and secondary helical strands. Qin et al. [18] established a numerical model for superconducting cables, analyzed the impact of the twist pitch on

the mechanical performance of filaments under electromagnetic forces, axial forces, and torsional moments, as well as the ultimate strain distribution. Wang et al. [19] established a multi-filament torsional finite element model for superconducting strands. Under cyclic loading, they analyzed the initial thermal residual stress and fracture of the conductor strands and their impact on the mechanical behavior. Xia [20] determined the relative positional relationship and geometric characteristics of the shearer cable structures, and the stranding exhibited an ideal tangent relationship. The conductors were simplified and treated as the minimum modeling unit. Solid modeling and assembly were completed through commands such as stretching, scanning, stretch removal, and scan removal. Simulated the overall tension of the cable, simulated the bending of the cable using the three-point bending method, and studied the effect of the cable pitch diameter ratio on cable tension and bending. The loads in the above studies are mostly static or exhibit small periodic deformations, and the research scales and angles are relatively singular.

Shearer cables undergo large deformations and endure harsh working conditions during their frequent back-and-forth movements. The development of coal mining towards intelligence and unmanned operations imposes higher demands on its comprehensive mechanical performance and fatigue life. This paper addresses problems in the parametric modeling of shearer cables, including determining tangency within and between conductor layers, recursively generating spiral curves from the (n-1)-th level to the n-th level, and constructing irregular surfaces for insulation and sheath. From the perspectives of tensile and bending properties, this study explores the influence of varying pitch diameter ratios at different stranding levels, stranding directions, and monofilament units on mechanical performance of shearer cables at the scale ranging from monofilaments to strands and from strands to conductors.

## 2. Parameterized modeling of shearer cables

Compared to traditional modeling approaches, parametric modeling has increased the efficiency of cable design. By discretizing the spiral curve with spatial points using interpolation, fitting the curve and surface in a lattice form, the method reduces the risks of errors and interference by eliminating assembly issues. This process significantly improves model accuracy, rendering the generated models more suitable for finite element analysis. In contrast to parametric modeling relying on the secondary development of CAD software, Grasshopper stands out as a visual node-based programming plugin integrated into the Rhino platform. Equipped with various operators to streamline cable design, Grasshopper also allows the development of new functional modules through Python and C#.

Taking MCPT-1.9/3.3KV shearer cable as the subject of engineering research, an accurate three-dimensional model is established using Rhino Grasshopper. After rendering by V-Ray, the cross-sectional structure and sectional view are illustrated in Fig 1. Compared to prior research on modeling of shearer cables, this paper presents a parameterized modeling approach for shearer cables utilizing Rhino Grasshopper, offering enhanced modeling accuracy and efficiency compared to prior research. As shown in Fig 2, Unlike the oversimplified model, this method preserves more geometric structural intricacies of the cables. It employs visual programming for constructing recursion from (n-1)-th to n-th levels, eliminating the necessity of calculating intricate high-order spiral curves and assembly, thus facilitating convenient parameter adjustments. Additionally, it conducts quantitative assessments of rubber sheath and insulation deformation during extrusion and cable formation, enabling precise establishment of irregular curved surface structures.

With the strand as the smallest analytical unit, the ground conductor is the first-level stranding structure, the power conductor is the second-level multi-layer stranding structure,

(A)

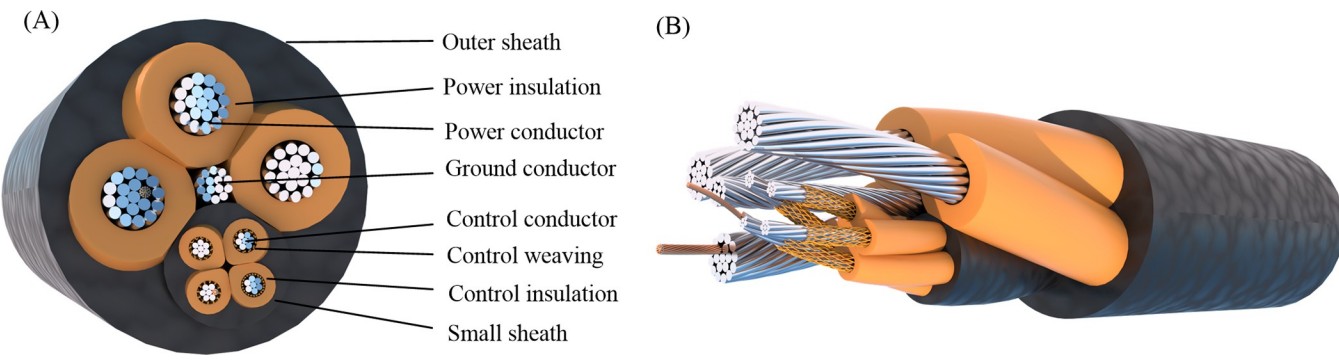

Fig 1. Shearer cable structure: (A) cross-section view; (B) anatomical view.

and the control conductor is the third-level stranding structure. The ground conductor is positioned at the center of the cable. The control conductor, serving as the fourth core, is combined with three power conductors to compose the shearer cable.

## 2.1 Tangency determination within and between layers and recursion from (*n*-1)-th level to *n*-th level

Parametric modeling of hierarchical stranding structures requires ensuring non-interference and maintaining the relative density of the structure throughout the processes of modeling from individual filaments to strands, strands to conductors, and ultimately to cable formation. Therefore, it is necessary to determine the tangency within and between layers of hierarchical stranding structures.

In the Frenet frame, at any point along the curve, the normal plane undergoes changes and presses toward the center of the structure, resulting in an elliptical shape for the section projection. The unfolded cylindrical helix is shown in Fig 3, where the stranding radius is defined as $R$, $D$ is the stranding outer diameter, and $j$ is the pitch. Lay angle is defined as $\beta$, and the helix

(A)                                                                      (B)

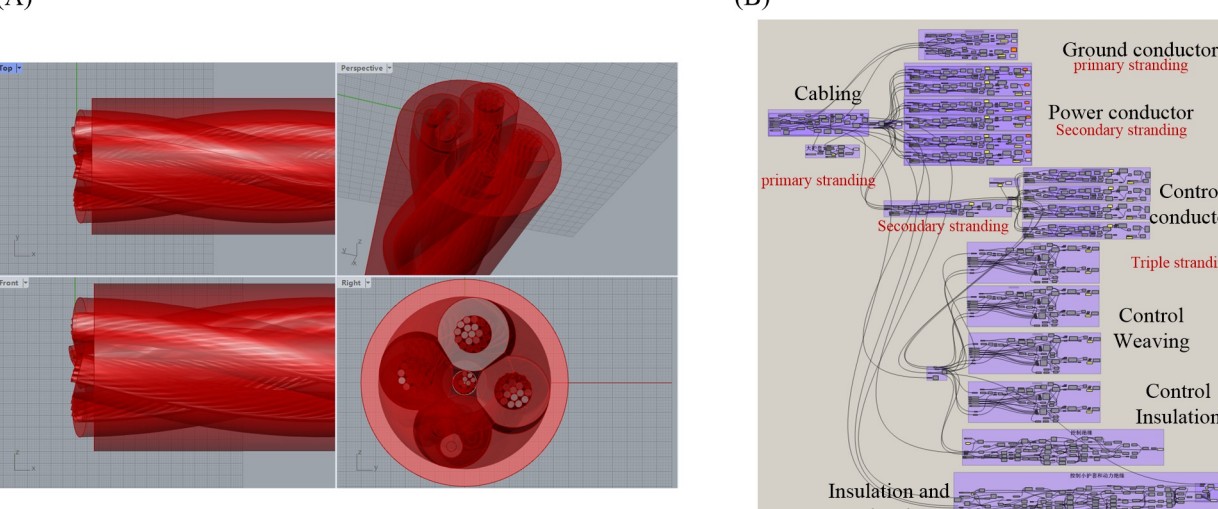

Fig 2. Parameterized modeling of shearer cables based on Rihno Grasshopper: (A) display of 3D models on the Rihno interface; (B) Visual programming in the Grasshopper interface.

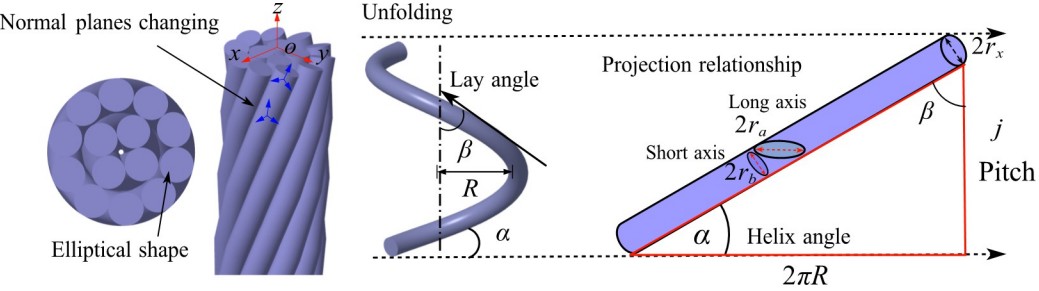

**Fig 3. Helical curve unfolding diagram.**

pitch angle $\alpha$ can be expressed by Eq (1). The semi-short axis $r_b$ is equal to the monofilament radius $r_x$. According to the projection relationship, the semi-long axis $r_a$ can be expressed by Eq (2).

$$\tan\alpha = \frac{j}{2\pi R} \tag{1}$$

$$r_a = \frac{r_x}{\sin\left(\arctan\left(\frac{j}{2\pi R}\right)\right)} \tag{2}$$

Fig 4A illustrates the within-layer tangency, where $R_s$ represents the radius of the strand at the intra-layer tangency. The elliptical tangent point $S(x, y)$ satisfies the standard elliptical Eq (3). The tangent slopes are determined through the inference from the elliptical tangent point and the tangent of the angle $\theta$, as expressed in Eq (4). Eq (5) represents the tangent line in point-slope form. The pitch diameter ratio $p$ is the ratio of the pitch to the stranded outer diameter, expressed by Eq (6). The joint Eqs (2)–(6) with 5 unknowns and 5 equations can be solved for a unique real number solution.

$$\frac{x^2}{r_a^2} + \frac{y^2}{r_x^2} = 1 \tag{3}$$

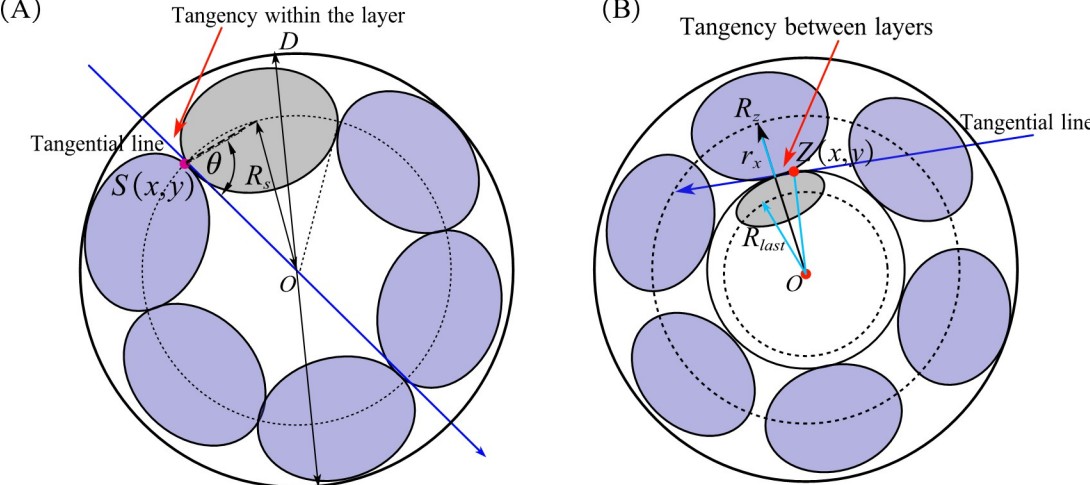

**Fig 4.** Determination of intra and inter layers tangency:(A) tangency within the layer; (B) tangency between layers.

$$-\frac{b^2 x}{a^2 y} = \tan(\frac{\pi}{2} - \frac{\pi}{n}) \tag{4}$$

$$y = \tan(\frac{\pi}{2} - \frac{\pi}{n})x - R_s \tag{5}$$

$$p = \frac{j}{2(R_s + r_x)} \tag{6}$$

Fig 4B illustrates between-layer tangency, where $R_z$ represents the radius of the strand at the inter-layer tangency, and $Z(x, y)$ is a point on the ellipse of the upper layer monofilament satisfying the elliptic equation. During inter-layer tangency, the distance $OZ$ from the point $Z$ on the ellipse of the upper layer to the center of the upper stranding attains its maximum value, expressed by Eq (7). The stranding radius $R_z$ can be defined as the sum of the monofilament radius $r_x$ and the maximum value of $OZ$, serving as the criterion for determining inter-layer tangency, as shown in Eq (8).

$$|OZ|\max = \sqrt{r_a^2 + (y + R_{last})^2 - (r_a/r_x)^2 y^2} \tag{7}$$

$$R_z = r_x + |OZ|\max \tag{8}$$

Utilizing Matlab for algorithm programming based on Eqs (2)–(8), by inputting arrays: total number of layers $C$, monofilament radius $r_x$, pitch diameter ratio $p$, and number of conductors in each layer $n$, it is possible to accurately and rapidly determine the tangency within and between layers for various hierarchical structures. Simultaneously, the algorithm calculates the stranding radius $R$ and pitch $j$ for each layer. The flowchart of the determination is illustrated in Fig 5.

In the Frenet-Serret frame [21], the centerline of the $n$-th level spiral curve aligns with the $(n\text{-}1)$-th level spiral curve. The construction entails creating the normal vector N to ensure its direction consistently aligns with the centerline, thereby constraining the increasing nature of the cosine and sine rotation angles. The normal vector N and the tangent vector T of the $(n\text{-}1)$ th level spiral curve together constitute the tangent plane. A normal vector B is derived through coordinate transformations, forming the normal plane represented by the vector BN.

$$(u, v) = (R_n\cos(\frac{2\pi l_{n-1}}{j_n}), R_n\sin(\frac{2\pi l_{n-1}}{j_n})) \tag{9}$$

The $n$ th-level pitch is defined as $j_n$, the stranding radius as $R_n$, and the arc length of the $(n\text{-}1)$ th-level stranding curve as $l_{n-1}$. The coordinates of the discrete points $(u, v)$ on the normal plane satisfy Eq (9). Taking advantage of Grasshopper's visual node-based programming, this recursive formula efficiently describes the transition from the $(n\text{-}1)$th-level stranding curve to the $n$th-level stranding curve, eliminating the necessity for deriving complex equations for higher-level spiral curves ($n \geq 3$).

## 2.2 Construction of shearer cable model

Quantitatively evaluate the deformations occurring in the sheath and insulation throughout the cable formation process, while precisely establishing their irregular surface structures. As

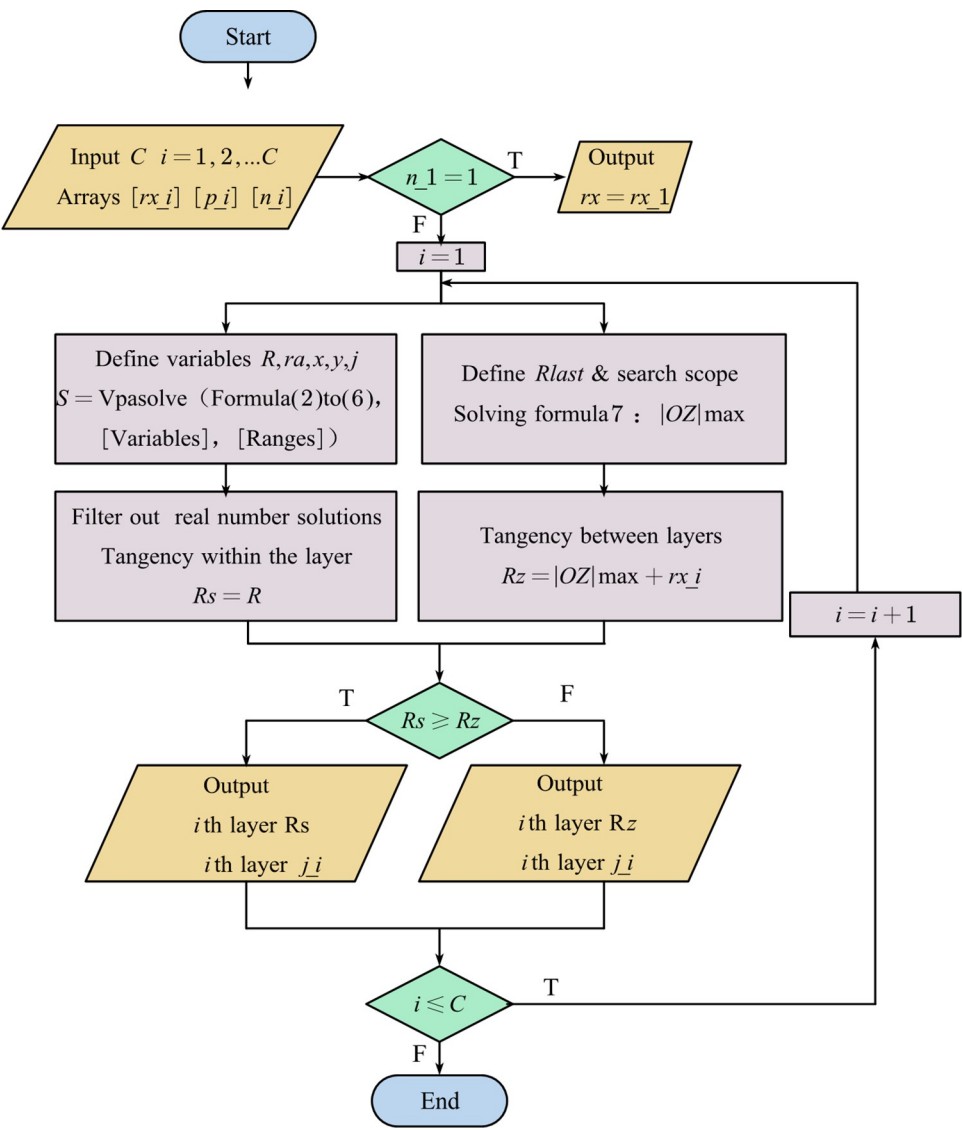

**Fig 5. Flowchart for the determination of intra-layer and inter-layer tangency.**

shown in Fig 6A, using the control insulation as an illustrative case, eight tangent curves are derived through extrusion deformation post-stranding, delineating the boundaries of the irregular surface. In Fig 6B, the irregular surfaces are filled with segments extending from the perpendicular projection of the centerline to discrete points on the corresponding curve. Chamfering is applied to regions with sharp angle connections. The fitting and sweeping process is applied to both inner and outer irregular surfaces, as depicted in Fig 6C and 6D. This approach facilitates the parametric modeling of control insulation, power insulation, small sheath of the control conductor, and outer sheath, as demonstrated in Fig 7.

Investigating the influence of pitch diameter ratios at different stranding levels, various stranding directions, and distinct filament units on the comprehensive mechanical performance of shearer cables in terms of tensile and bending properties. This exploration spans from individual filaments to strands and from strands to conductors. Taking advantage of the parametric modeling capabilities of Rhino Grasshopper, cable models with diverse parameter

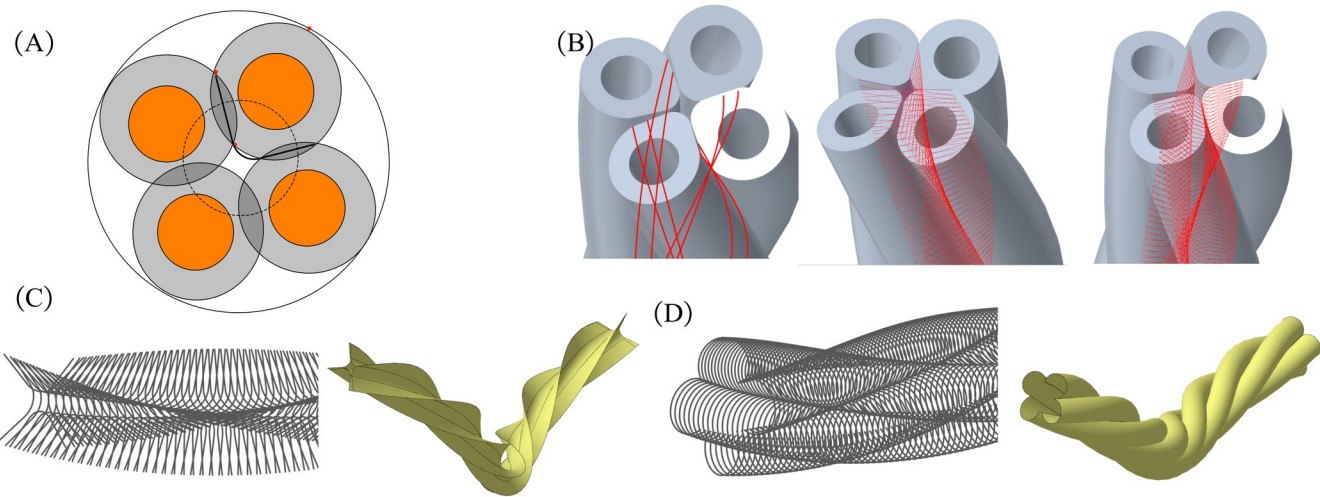

**Fig 6.** Construction of irregular surfaces: (A) cross-section; (B) generate tangent curves and fill surfaces; (C) fit and sweep inner surfaces; (D) fit and sweep outer surfaces.

combinations are created to facilitate numerical simulations. Fig 8 depicts cable models with varying pitch diameter ratios, Fig 9 showcases conductor models with different stranding directions, and Fig 10 presents conductors with distinct filament units.

## 3. Calibration of equivalent mechanical parameters using tensile test

The tensile test method was used for calibrating the equivalent mechanical parameters of insulation, sheath, and conductor strands. EPDM rubber is utilized for power insulation and control insulation, high-performance CPE rubber serves as the material for the small sheath of the control conductor, and neoprene rubber is used for the outer sheath. Fig 11A illustrates the

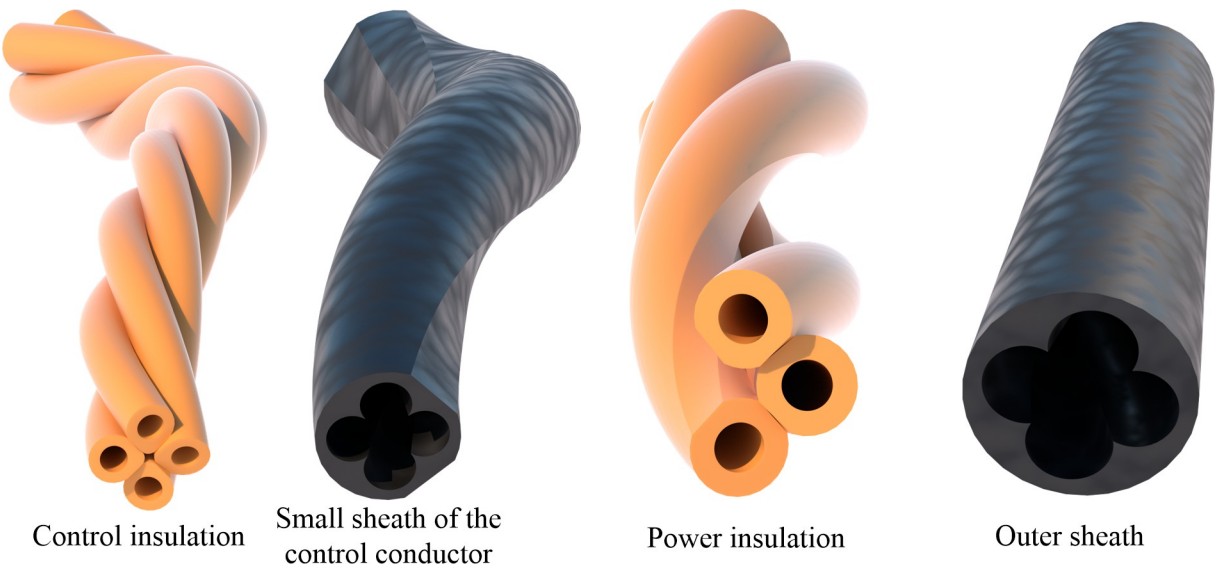

| Control insulation | Small sheath of the control conductor | Power insulation | Outer sheath |

**Fig 7. Models of insulation and sheath.**

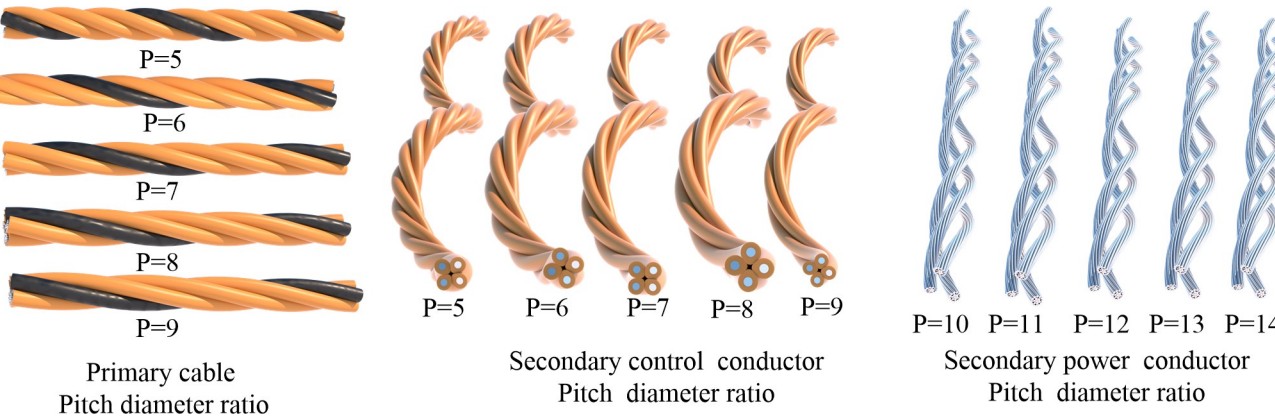

**Fig 8. Cable models with varying pitch diameter ratios at different stranding levels.**

preparation of dumbbell-shaped rubber specimens using a punching machine. According to the national standard GB/T 2951.11–2008, the movement speed of the clamp in the tensile test of rubber insulation and sheath is (250±50) mm/min, and the tensile speed of YN21003 was set to 200 mm/min. Subsequent to linear fitting, the equivalent elastic modulus, tensile strength, elongation at break, and tear strength are calibrated, as detailed in Table 1. The abbreviations AVG and SD in Tables 1 and 2 represent the mean and standard deviation.

The strands are formed by stranding filaments. Based on the homogenization theory and volume average principle, tensile tests were conducted to equivalently characterize the mechanical properties of the entire strand. In compliance with national standards GB/T 4909.3–2009 and GB/T 228–2002, the tensile speed of the AI-7000-LA20 tensile testing machine was set to 50 mm/min. As depicted in Fig 11B, the measured tension and displacement data were then converted into stress and strain curves, as demonstrated in Fig 12. Linear fitting within the elastic range is performed, and the 2% offset theory is applied to determine the yield strength [22]. The slope between yield strength and tensile strength is utilized to determine the tangent modulus, calibrating the equivalent mechanical parameters of the strands as presented in Table 2.

In subsequent numerical simulations, the bilinear isotropic hardening model is used to define the material parameters of the strands. Fig 13 illustrates the simulation of the material's elastic-plastic deformation, employing two linear segments. One segment depicts elastic behavior with a slope equivalent to the elastic modulus, while the other signifies plastic behavior with a slope equivalent to the tangent modulus. This methodology effectively captures the material's nonlinear traits while optimizing simulation efficiency. Defining the Bilinear

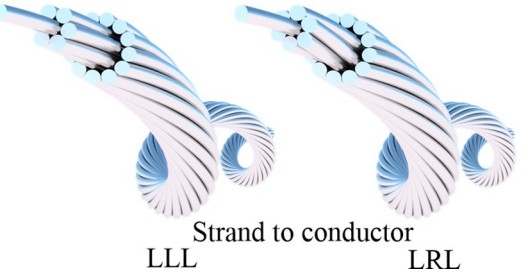

**Fig 9. Conductor models with different stranding directions.**

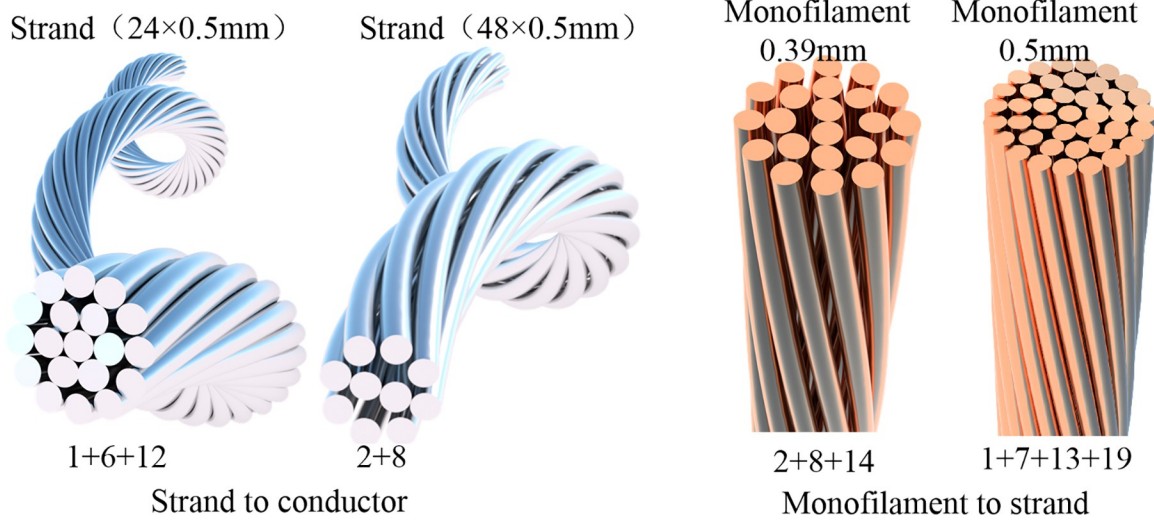

**Fig 10. Conductors with distinct monofilament units.**

Isotropic Hardening within finite element software material libraries entails specifying parameters including elastic modulus, yield strength, tangent modulus, and tensile strength.

## 4. Mechanical characteristics of shearer cables

### 4.1 Impact of pitch diameter ratios at various stranding levels on mechanical performance of shearer cables

The shearer cables undergo frequent short-distance back-and-forth movements while following the shearer, experiencing frequent bending, dragging, compression, twisting, and potential impacts from coal or falling rocks. The large deformations generated could lead to mechanical damage, thereby affecting the normal operation and safety of the shearer.

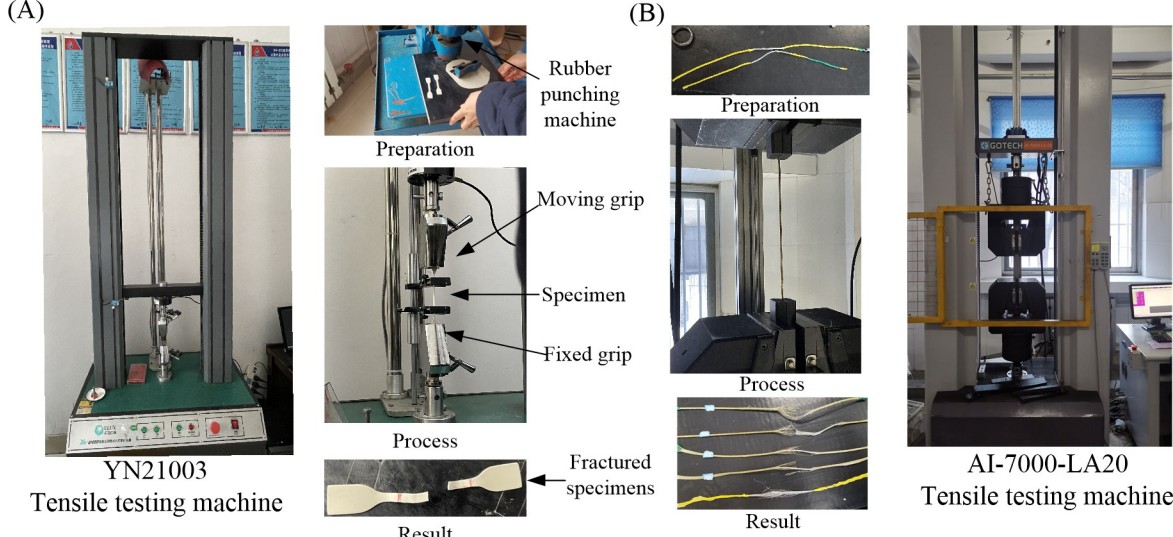

**Fig 11.** Tensile test: (A) rubber insulation and sheath; (B) tinned copper strands.

**Table 1. Mechanical parameters of insulation and sheath.**

| Material (rubber) | Elastic modulus (MPa) | | Tensile strength (MPa) | | Elongation (%) | | Tear strength (MPa) | |
|---|---|---|---|---|---|---|---|---|
| | AVG | SD | AVG | SD | AVG | SD | AVG | SD |
| EPDM | 3.27 | 0.87 | 12.02 | 0.51 | 370.08 | 17.59 | 48.04 | 2.39 |
| CPE | 3.81 | 0.83 | 14.58 | 0.47 | 407.62 | 27.84 | 58.32 | 1.92 |
| Neoprene | 3.85 | 0.79 | 16.02 | 0.74 | 362.25 | 28.62 | 62.38 | 3.76 |

As shown in Fig 14, define the direction of increasing bending radius $R$ as the positive strain direction, and $\theta$ as the arc angle. During the cable bending process, the length of arc $AB$ on the neutral layer remains unchanged. On the outer side of the bend, the length of arc $CD$ undergoes tensile elongation, while on the inner side of the bend, the length of arc $EF$ undergoes compressive shortening. The strain $\varepsilon$ can be derived from Eq (10).

$$\varepsilon = \frac{CD - AB}{AB} = \frac{(R+y)\theta - R\theta}{R\theta} = \frac{y}{R} \tag{10}$$

Within the elastic range, satisfying Hooke's Law, the bending stress $\sigma$ is obtained from Eq (11), where $E$ is the elastic modulus:

$$\sigma = E\varepsilon = \frac{Ey}{R} \tag{11}$$

$A$ is the cross-sectional area, $d$ is the diameter of the cable, and the area moment of inertia $I$ for the cylinder is expressed by Eq (12):

$$I = \frac{\pi d^4}{64} \tag{12}$$

The internal resultant moment in the cable is the bending moment $M$, and which can be derived through integration, as expressed in Eq (13). The bending stress $\sigma$ can also be represented by Eq (14):

$$M = \int_A \sigma y \, dA = \frac{E}{R} \int_A y^2 \, dA = \frac{EI}{R} \tag{13}$$

$$\sigma = \frac{My}{I} \tag{14}$$

**Table 2. Mechanical parameters of strands.**

| Strand (Tinned Copper) | Elastic modulus (GPa) | | Yield strength (MPa) | | Tangent modulus (MPa) | | Tensile strength (MPa) | | Elongation (%) | |
|---|---|---|---|---|---|---|---|---|---|---|
| | AVG | SD | AVG | SD | AVG | SD | AVG | SD | AVG | SD |
| Control KA6 | 16.43 | 0.97 | 160.34 | 8.62 | 506.70 | 19.89 | 238.04 | 2.35 | 28.53 | 0.80 |
| Power 24×0.5 | 37.16 | 2.13 | 233.41 | 19.89 | 805.72 | 30.13 | 270.39 | 3.62 | 23.60 | 3.33 |
| Power 48×0.5 | 17.86 | 0.57 | 182.25 | 5.15 | 234.38 | 12.63 | 257.25 | 0.63 | 28.85 | 1.48 |

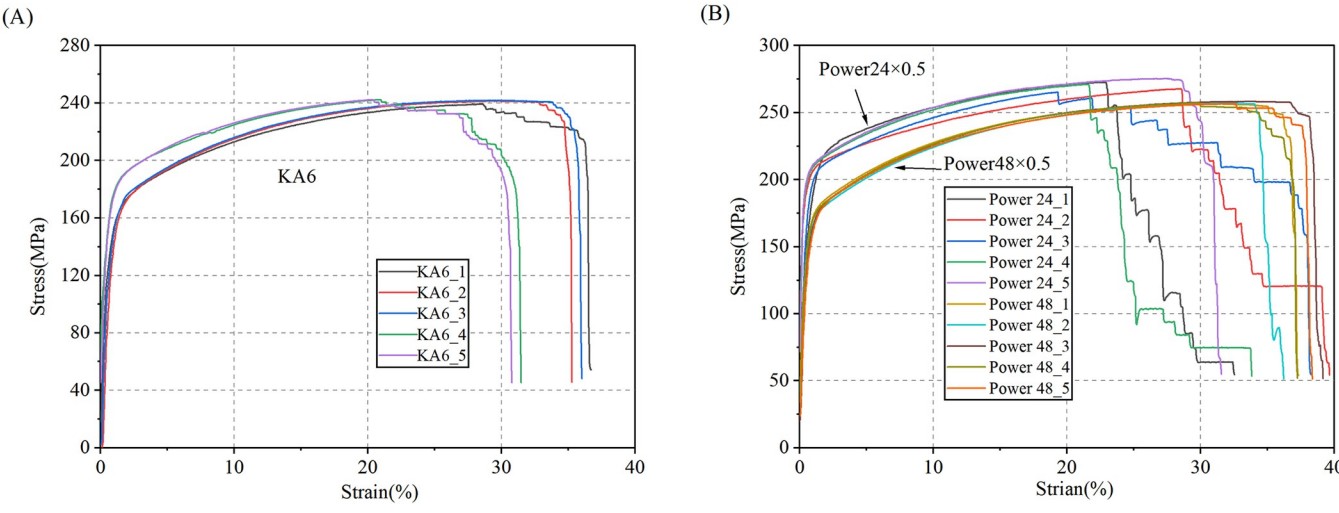

**Fig 12.** Stress -strain curves of strands: (A) control strands; (B) power strands.

Bending stress increases linearly with bending moment $M$ and distance from the neutral axis $y$. The maximum stress occurs on the arc length furthest from the neutral axis.

$$R = \frac{L}{2\tan\left(\frac{\alpha}{2}\right)} \tag{15}$$

Applying a bending load to the cable by rotating the cable clamp, the cable reaches its minimum bending radius when the cable clamp rotation angle approaches the limit. In the bending test of shearer cables [23], the minimum bending radius of the cable is limited by the

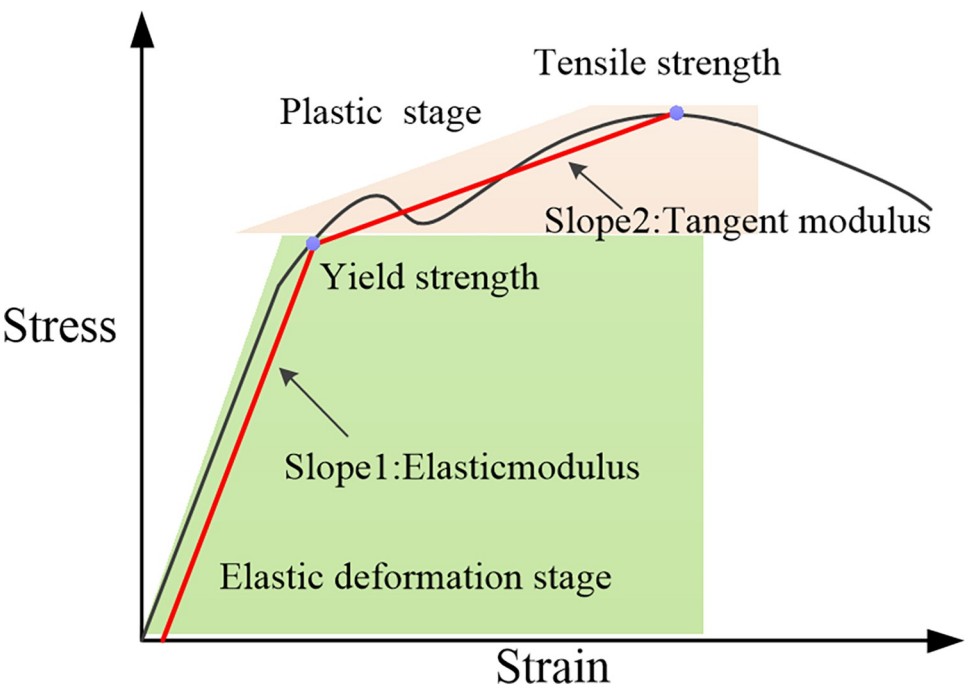

**Fig 13. Schematic diagram of bilinear isotropic hardening model.**

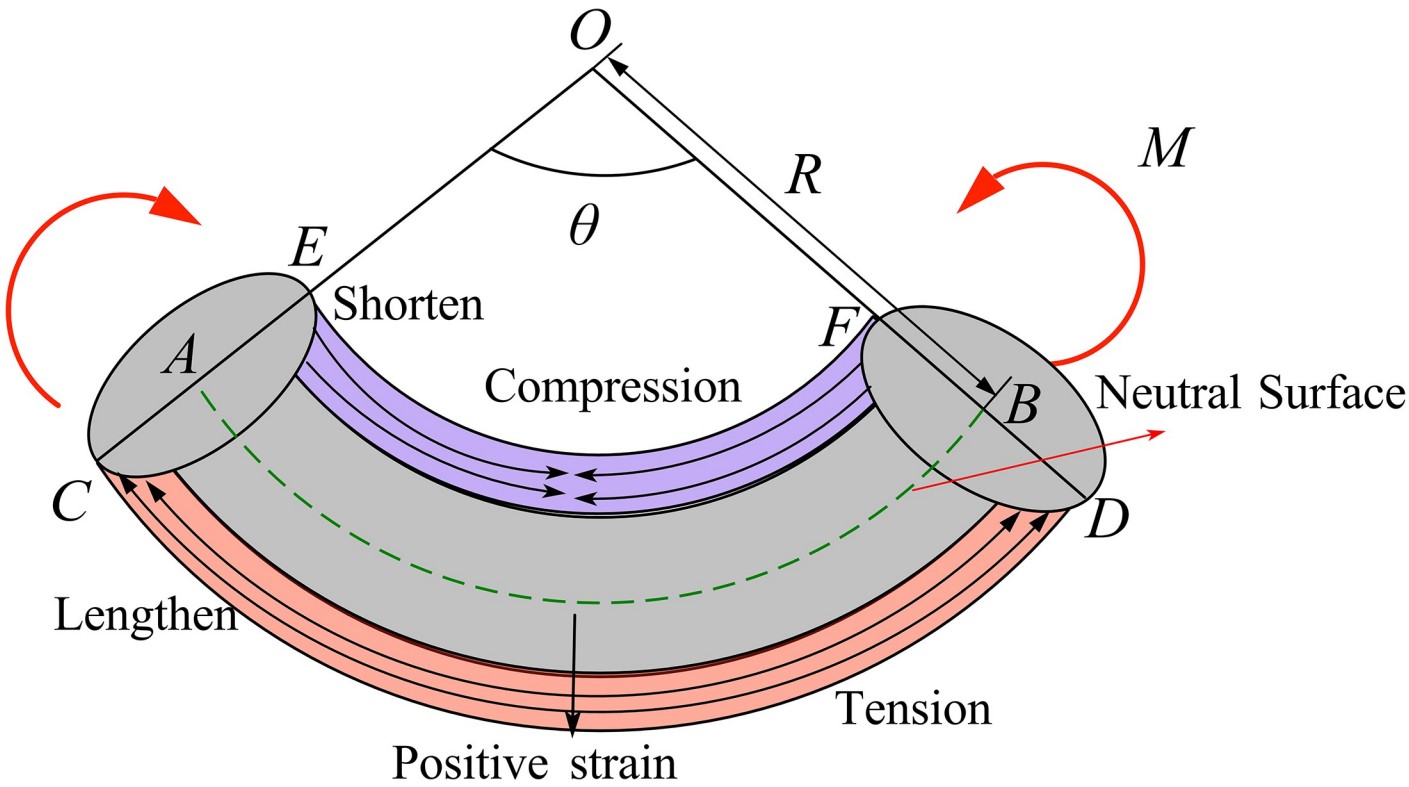

**Fig 14. The mechanical model of cable bending.**

geometrical parameters of the cable clamps. As shown in Fig 15, the minimum bending radius $R$ can be calculated from the limiting bending angle $\alpha$ and the pin hole centre distance $L$, as shown in Eq (15), based on the inference of the inner angle of the positive N-side shape, the length of the side, and the radius of the inner joint circle. According to the bending test standard for shearer cables GB/T12972.1–2008, the cross-sectional area of is 95 mm², the pin hole centre distance $L$ is selected as 100 mm, the maximum bending angle $\alpha$ is 30 degrees, and the cable's bending radius $R$ is 200 mm.

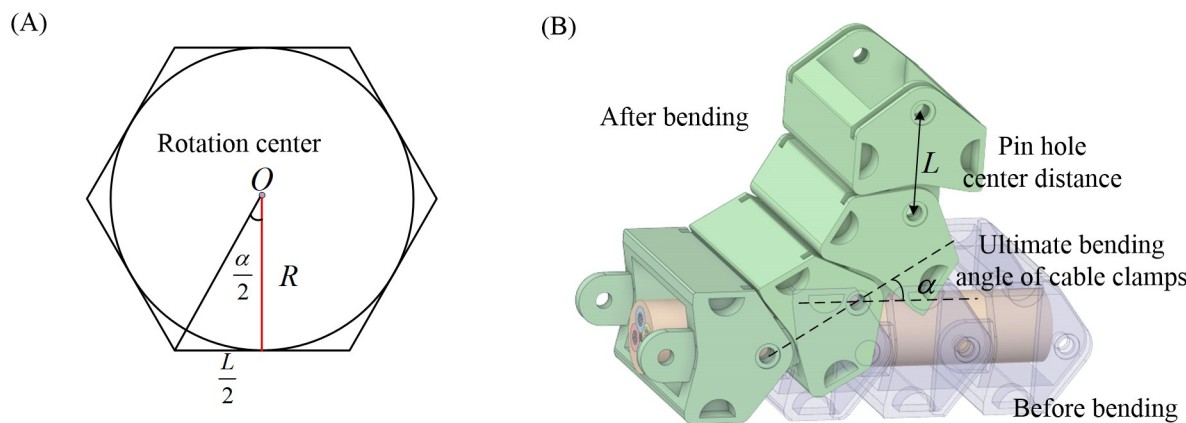

**Fig 15.** Bending simulation schematic of the shearer cable: (A) geometric diagram; (B) process of bending.

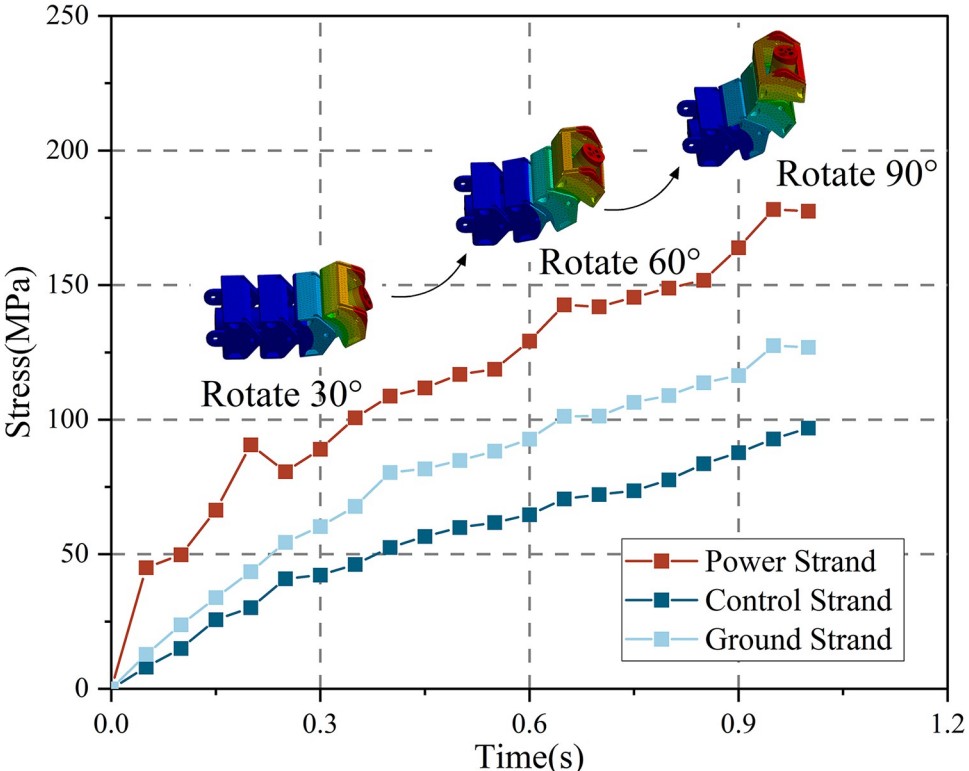

**Fig 16. Simulation process and stress changes of strand.**

The finite element software ANSYS LS-DYNA is used for explicit dynamic analysis. The simulated cable was intercepted to 400 mm to guarantee the inclusion of at least one pitch characteristic. A rotational center coordinate system was defined, and the six degrees of freedom of the initial cable clamps were constrained with remote displacement. Additionally, revolute joints were introduced between adjacent cable clamps, enabling the end cable clamps to rotate around the rotation center by 90 degrees, thereby accomplishing a rotational increment of 30 degrees for each cable clamp.

As depicted in Fig 16, the complete bending process is segmented into three phases: 0 to 0.33 s, 0.33 to 0.66 s, and 0.66 to 1 s. Each cable clamp systematically attains a rotational increment of 30 degrees, reaching its maximum angle, while each cable segment reaches the limit bending radius. Over the course of the bending process, stress levels progressively rise with time, and the motion process is in accordance with the loading of the bending test.

The calibrated equivalent elastic modulus for insulation, small sheath, and outer sheath in tensile testing are 3.27 MPa, 3.87 MPa, and 3.85 MPa, respectively. As shown in Fig 17, compared to tinned copper, the rubber material exhibits a lower elastic modulus. The maximum equivalent stress during the bending process ranges from 0.76 to 3.1 MPa, with stress levels below the rubber material's strength limit of 10 MPa. This effectively safeguards the conductor, making the cable better suited for the complex and frequent reciprocating bending movements in the coal mine.

The nephogram depicting the maximum equivalent stress of the conductors is presented in Fig 18. The power conductor and control conductor are secondary stranding structures, positioned away from the neutral layer and periodically distributed on the inner and outer sides of the bending cable. The outer side experiences tension, while the inner side undergoes

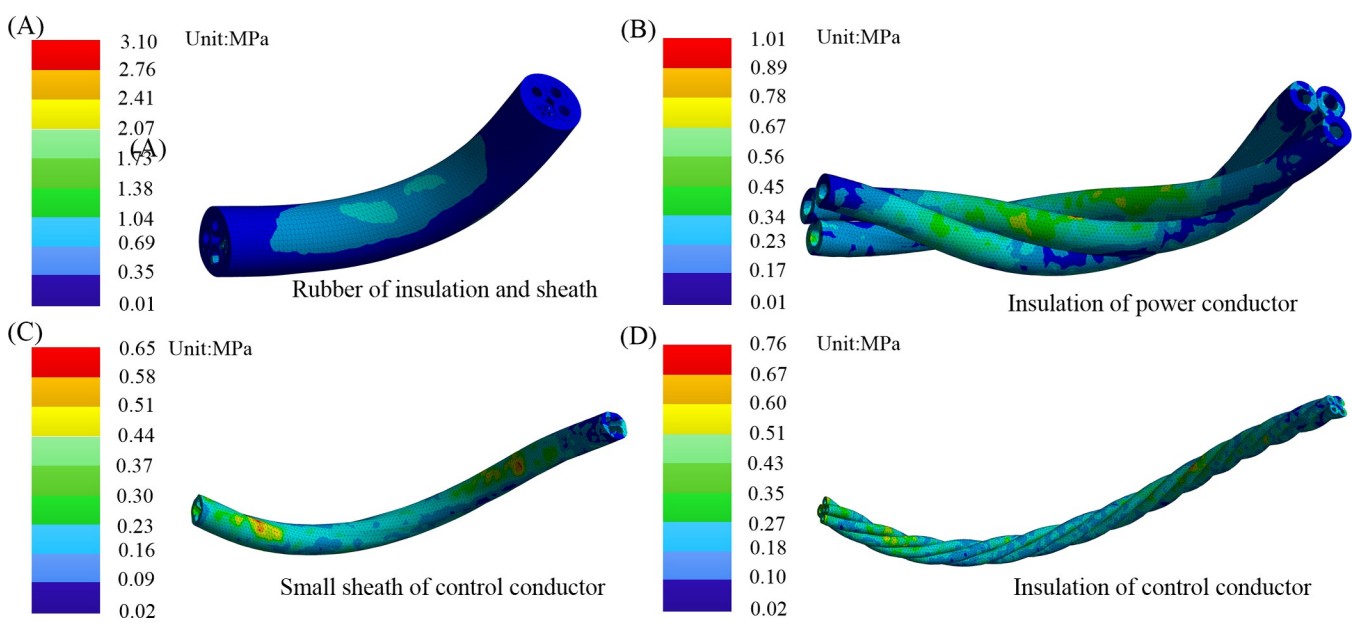

**Fig 17.** Maximum equivalent stress nephogram: (A) whole rubber; (B) power insulation; (C) small sheath of control; (D) Outer sheath.

compression. During the bending process, the maximum equivalent stress for the power conductors is 183.04 MPa, and for the control conductors, it is 120.01 MPa. The yield strength and tensile strength of the control strands are 160.34 MPa and 238.04 MPa, while those of the power strands are 233.41 MPa and 270.39 MPa, respectively. The analysis of the conductors focuses primarily on the power conductor and control conductor, both of which produce cumulative fatigue damage. Further fatigue life analysis is necessary. While the ground conductor, constituting a primary stranded structure, is located at the center of the cable structure, proximate to the neutral layer region. Upon reaching the limit bending angle, it exhibits a maximum equivalent stress of 96.84 MPa. Its fatigue strength is 135.195 MPa,so it does not incur cumulative damage.

Utilizing the loading and displacement method depicted in Fig 16, successive cable clamps achieved a 30-degree incremental rotation, culminating in the cable reaching the ultimate bending state depicted in Fig 19A. This study examines the stress distribution along the axial arc length of the cable, employing the ideal simply supported beam model illustrated in Fig 19B. When the cable is subjected to the ultimate bending state, the inner wall of the cable clamp applies a uniformly distributed load $q$ along the axial length $l$ of the cable. Applying the equilibrium equation, shear forces are opposite at both ends, reaching a maximum of $ql/2$. At the midpoint, the shear force diminishes to zero, while the bending moment reaches its maximum value at the midpoint, $ql^2/8$, with both ends registering zero. Additionally, since stress magnitude is directly proportional to the bending moment, stress distribution in the ultimate bending state of cables adheres to the pattern of being greater in the middle and smaller at both ends along the arc length. In the simulation results, stress paths along the arc length are established for the power, control, and ground strands on the outer side of conductors, as shown in Fig 19C. Stress distribution curves along the arc length are plotted, as depicted in Fig 19D. In comparison to the ideal mechanical model, the simulated stranded wire experiences more complex combined loads in the simulation, leading to fluctuations in the curves. Nonetheless, it also demonstrates the distribution pattern of higher stress levels in the middle and lower stress levels at both ends.

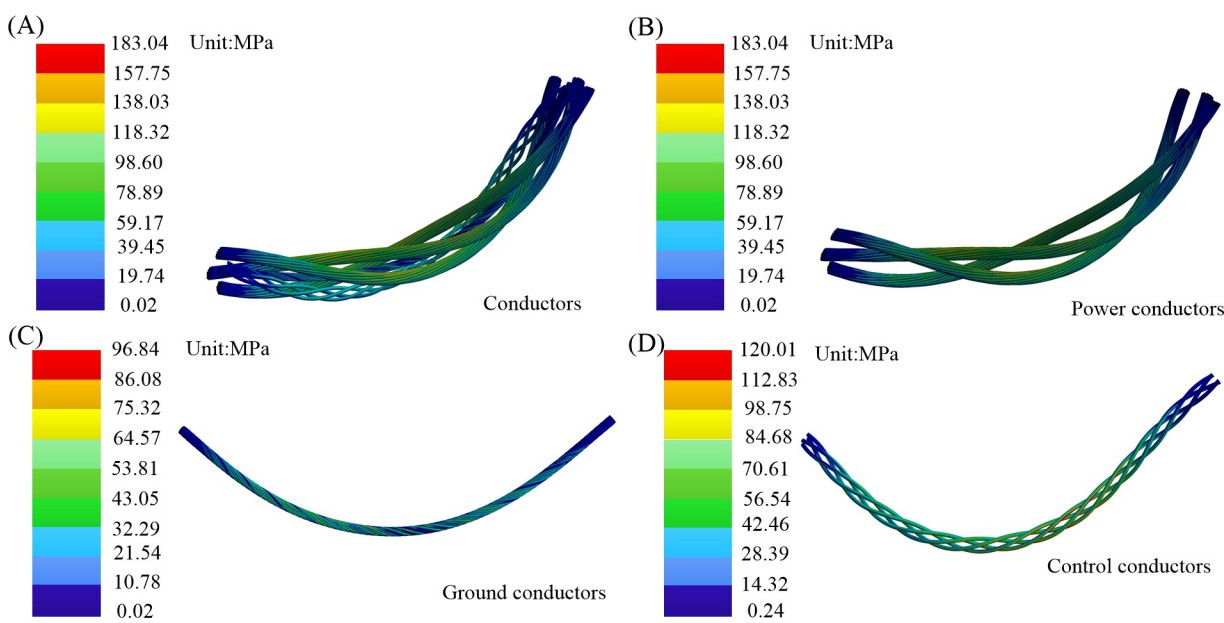

**Fig 18.** Maximum equivalent stress nephogram: (A) whole conductors; (B) power conductors; (C) ground conductors; (D) control conductors.

The study investigated the impact of pitch diameter ratios at various stranding levels on the mechanical performance of the cable under bending conditions. Scatter plots illustrating the maximum bending stress of power and control strands at different grade pitch diameter ratios are presented in Fig 20. Multiple linear and polynomial fits are conducted considering the characteristics of the data. Using Matlab software, trend nephograms depicting the maximum equivalent bending stress of power and control strands with varying pitch diameter ratios at different stranding levels are generated, as illustrated in Fig 21.

Maintaining a constant pitch diameter ratio for the secondary stranding of power and control conductors, when the pitch diameter ratios of the primary cabling stranding are set at 4, 5, and 6, the stresses in the cable are relatively close. The slopes of the fitting curves $y_1$ and $y_3$ are small, measuring 7.5 and 8, the coefficient of determination $r^2$ are 0.997 and 0.989 respectively. The stress levels for the power conductors are 168 MPa, 176 MPa, and 183 MPa, while for the control conductors, they are 106 MPa, 115 MPa, and 120 MPa. However, when the pitch diameter ratios of the primary cabling stranding are set at 7 and 8, the stress levels are higher, and the growth trend is evident. The slopes of the fitting curves $y_2$ and $y_4$ are larger, measuring 17.5 and 16.5, $r^2$ are 0.995 and 0.994, respectively. The stress levels for the power conductors reach 202 MPa and 218 MPa, while for the control conductors, they reach 131 MPa and 138 MPa.

Maintaining a constant pitch diameter ratio for primary cabling stranding, the linear fitting curve $y_5$ for the stress of power conductors with varying pitch diameter ratios at the secondary stranding has a very small slope of 0.306, and the coefficient of determination $r^2$ is 0.907. The power conductor is a multi-layer structure of the same stranding level. Within the given range, the trend of stress levels with pitch diameter ratios is not prominent. However, after performing a cubic polynomial fitting on the scatter plot of the stress of the control conductor, the resulting curve $y_6$ is represented as $0.75x^3+12x^2-66.25x+221$, and $r^2$ is 0.992. At pitch diameter ratios of 6, 7, and 8, the stress values for the control conductor are relatively high, reaching 131 MPa, 137 MPa, 140MPa, and approaching the yield strength. Meanwhile, for pitch diameter ratios of 4 and 5, the stress levels grow gradually and are lower, measuring 117 MPa and 120 MPa.

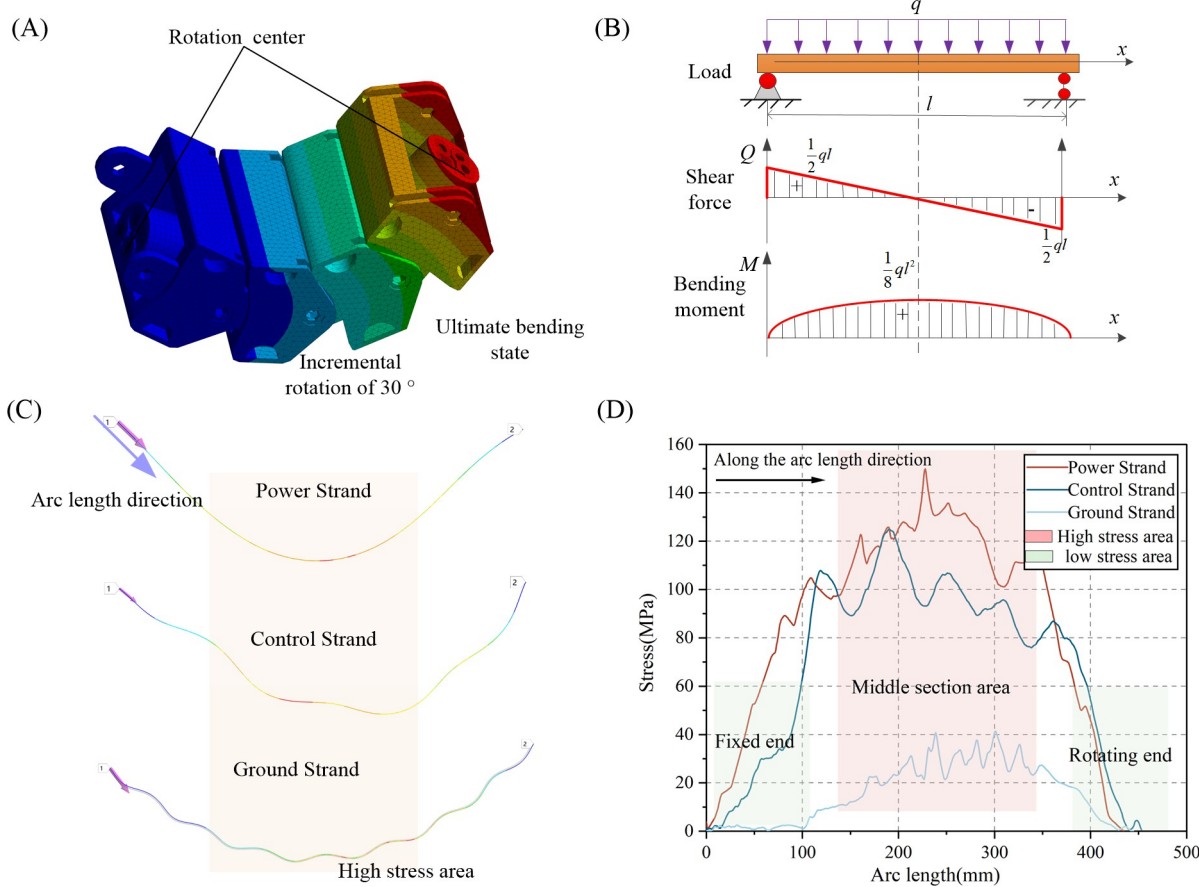

**Fig 19.** Stress distribution along the arc length direction: (A) schematic diagram of cable ultimate bending; (B) shear force and bending moment diagram of an ideal simply supported beam; (C) Stress nephogram along the arc length direction of strands; (D) Stress -arc length curves along the arc length direction of strands.

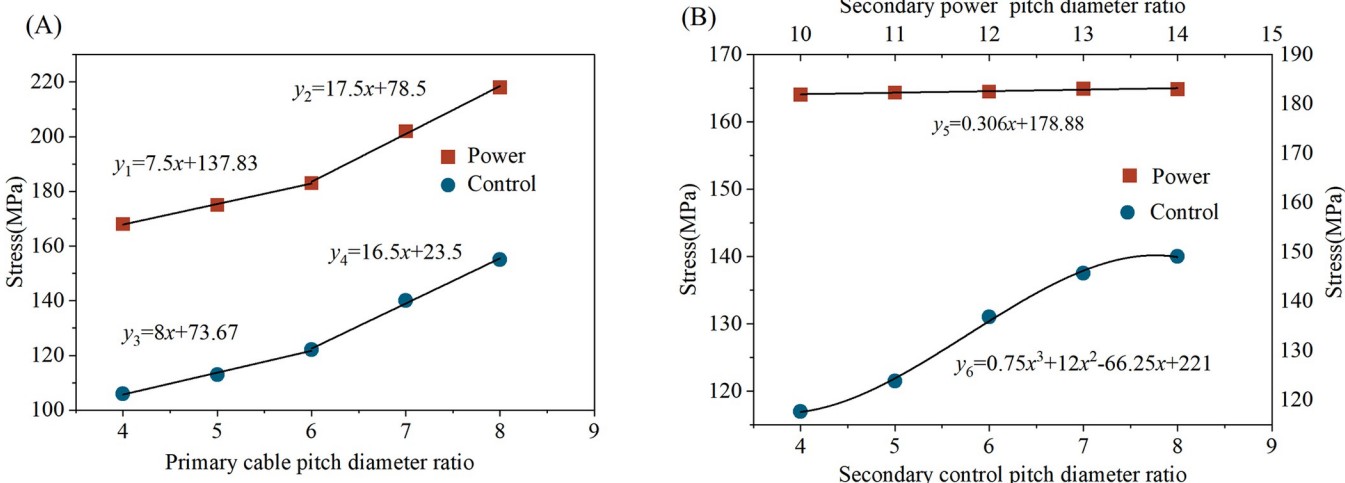

**Fig 20.** Scatter plots with fitted curves for maximum equivalent stress of strands: (A) Primary pitch diameter ratios; (B) Secondary pitch diameter ratios.

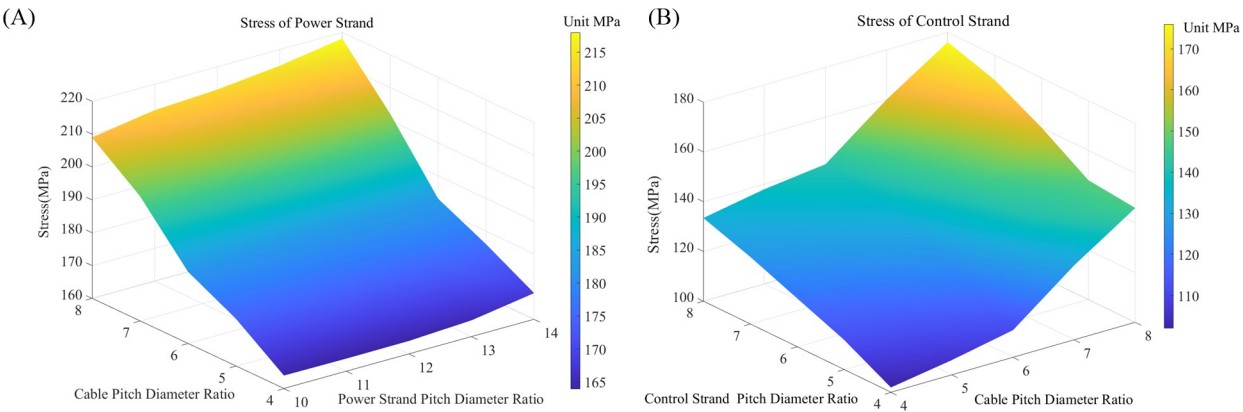

**Fig 21.** Trend nephograms of strands with varying pitch diameter ratios at different stranding levels:(A) power strands; (B) control strands.

Under the same axial length conditions, a smaller pitch diameter ratio of conductor results in greater copper consumption, leading to an increase in production costs. Therefore, while maintaining close stress levels, it is advisable to choose a pitch diameter ratio within a reasonable upper limit. Considering the influence of various pitch diameter ratios at different stranding levels on mechanical properties, the optimal design for the pitch diameter ratios of this shearer cable is as follows: select a pitch diameter ratio for primary cabling stranding as 6, for the power conductor at secondary stranding as 14, and for the control conductor at secondary stranding as 5.

Previous studies have explored the impact of primary cable stranding diameter ratios on cable tensile and bending performance, overlooking the holistic influence of secondary dynamic and control cable strand diameter ratios. The bending tests typically employed three-point bending or direct cable model bending, deviating from actual cable clamp-induced bending conditions and thus presenting limitations. This chapter investigates the impact of the pitch diameter ratio of different stranding levels on the comprehensive performance of shearer cables. The loading method adheres to bending test standards and mirrors real underground loading scenarios, demonstrating innovation and progress.

## 4.2 Fatigue life prediction of shearer cables and experimental verification

The stress amplitude and number of cycles satisfy a logarithmic function relationship [24, 25]. Based on engineering practice and preliminary finite element simulation, bending tests of the shearer cable with over $5e^4$ bending cycles classify as high-cycle fatigue. Additionally, stress values in the finite element simulation mostly fall below the yield strength, indicating the rationale behind studying stress fatigue strength. Concerning bending loads, stress range amplitude $S1$ for $10^3$ cycles corresponds to 90% of the tensile strength, while for $10^6$ cycles, it is 0.357 times the tensile strength. Parameter $b1$ represents the first fatigue strength exponent, defining the slope in the high-cycle fatigue segment on logarithmic coordinates, as illustrated in Formula (16). Given that $S2$ is smaller than $S1$, the slope is negative. Parameter $b2$ denotes the second fatigue strength exponent, derived from $b1$ using Formula (17). Parameter $SRI1$ indicates stress range intercept, describing the stress range at a cycle count of 1, as expressed by Formula (18). Mechanical tests were conducted on power and control strands to obtain a fitted S-N curve with modified ultimate tensile strength, as depicted in Fig 22.

$$b1 = \frac{\lg(S2) - \lg(S1)}{\lg(10^6) - \lg(10^3)} = \frac{\lg(\frac{S2}{S1})}{3} \quad (16)$$

$$b2 = \frac{b1}{2 + b1} \tag{17}$$

$$\frac{SRI1}{2} = \frac{S2}{(10^6)^{b1}} \tag{18}$$

A new cable model was constructed according to the optimal pitch diameter ratio scheme and new process was adapted to produce this new cable. Import the finite element simulation results of the shearer cable bending into Ncode software for fatigue life analysis [26]. The calculation and analysis process is illustrated in Fig 23. Utilizing finite element analysis results and time-step load spectra automatically generated by the software, stress history is derived without the necessity of calculating average stress or stress amplitude for load spectrum generation. This approach is applicable for transient and large deformation explicit finite element solution results. The fatigue life nephograms and fatigue damage values for both power and control strands can then be determined.

Fig 24 depicts the fatigue life nephograms for the strands. The power strands exhibit the minimum cycle count at node 612481, with 2.942e$^5$ cycles and a corresponding fatigue damage

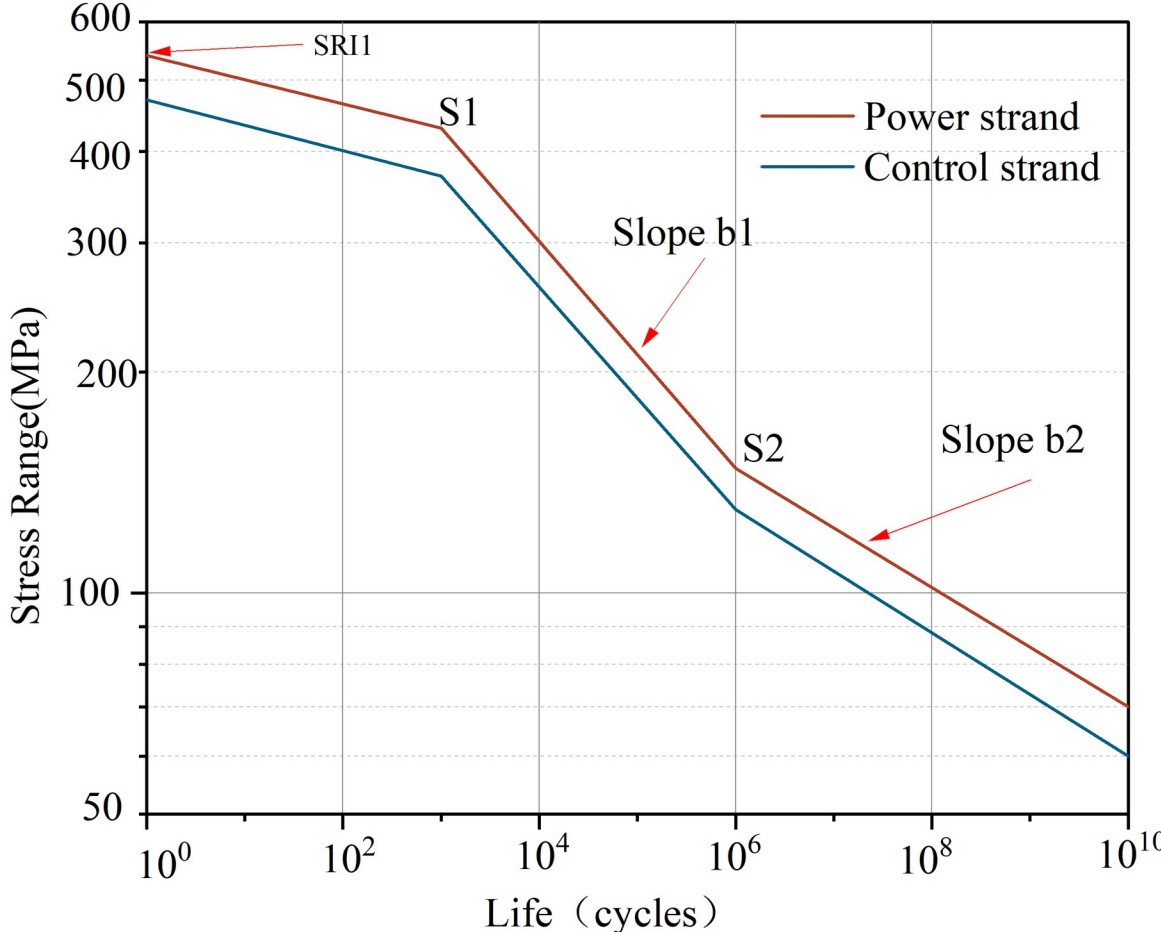

**Fig 22. S-N curve with modified ultimate tensile strength.**

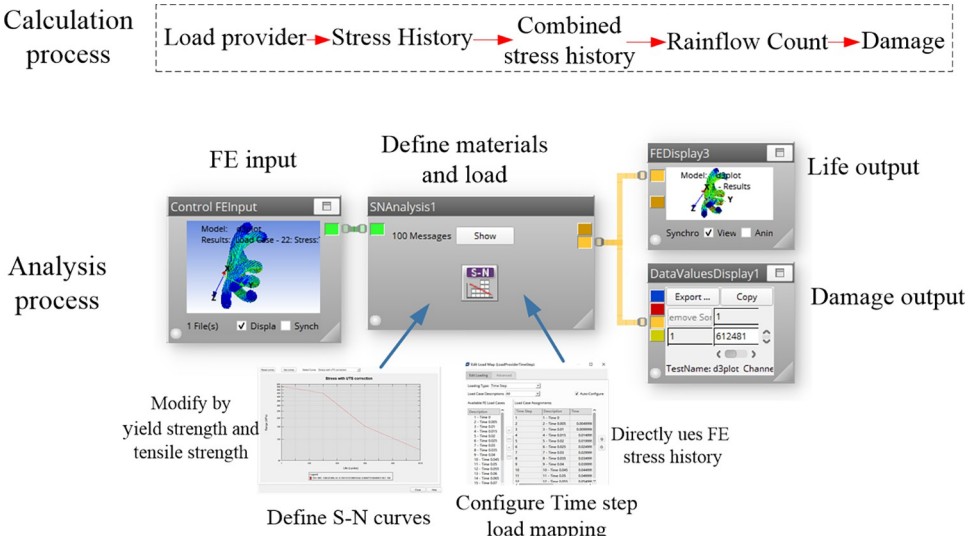

**Fig 23. Calculation and analysis process of stress fatigue life.**

value of 3.397e$^{-6}$. As for the control strands, the minimum cycle count occurs at node 20011, with 3.024e$^5$ cycles and a fatigue damage value of 2.205e$^{-6}$. During the stress analysis of cable bending, although the stress level of the power strands is higher than that of the control strands. However, when considering the fatigue life analysis based on the yield strength and modified S-N curves, the fatigue life and damage of the control strands are found to be close to those of the power strands, which is consistent with the practical production that the control conductor is susceptible to cumulative damage. Additionally, this also verifies that the research methodology can reasonably evaluate the effect of structural changes on the mechanical properties of shearer cables from both stress level and fatigue life prediction perspectives.

The fatigue life of the shearer cable before improvement is lower, reaching 7e$^4$ cycles, with slight serpentine and bulging observed during the primary cabling formation. Following the optimal pitch diameter ratio scheme, a new type of cable was produced through process adjustments and experimental trials. As shown in Fig 25, a cable sample with a length of 4500 mm

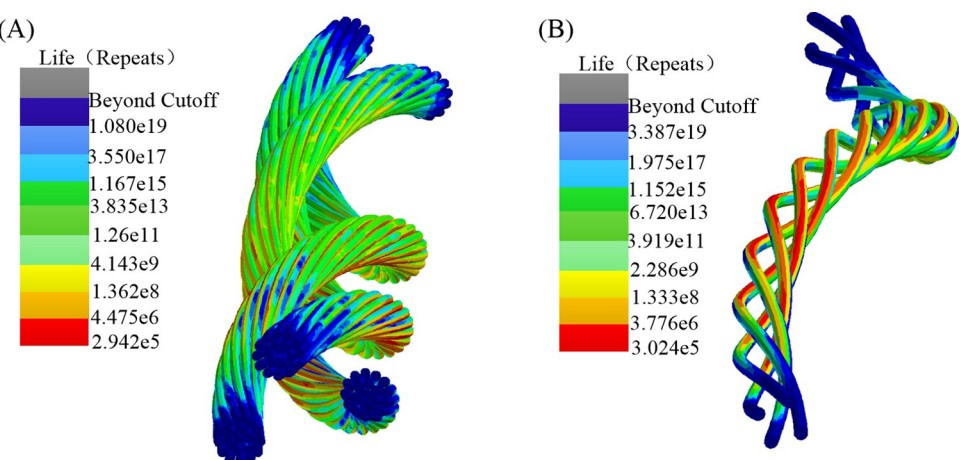

**Fig 24.** Fatigue life nephograms: (A) power strands; (B) control strands.

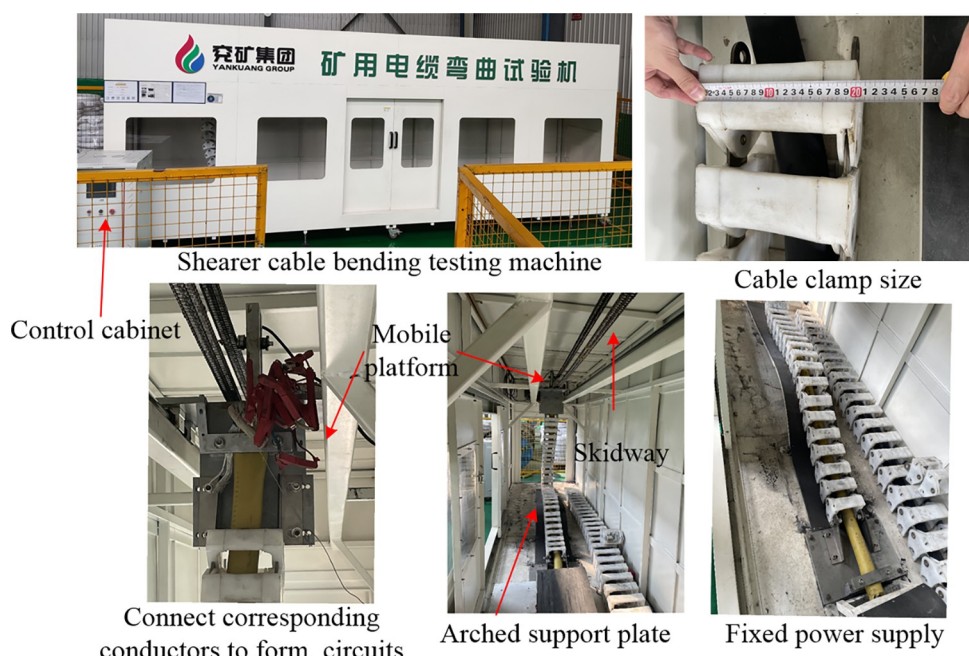

**Fig 25. Cable bending test machine.**

was threaded through cable clamps, and the sheath and insulation at both ends of the cable were stripped. One end was securely fastened to the sliding moving plate using fixtures, while the other end was fixed to the power supply. Each conductor was connected to form a circuit according to the corresponding markings, and the moving speed was set to 18 m/min. When a short circuit or open circuit occurs in the shearer cable, the control panel displays the corresponding fault, and a buzzer alarm sounds. The number of cable bends at this moment is taken as its maximum bending fatigue life.

The cable bending test spanned 40 days, during which the maximum number of cable bends reached $14.13e^4$ in the presence of short circuit and open circuit faults. One cycle of cable bending, involving both the bending and recovery of the cable, is defined by the reciprocating motion of the moving plate Fig 26A illustrates that in accordance with the coal industry standard MT 818.1–2009, the bending test machine subjects the tested cable section to a bending process transitioning from a straight to an S-shaped state during each reciprocating cycle. Conversely, as depicted in Fig 26B, the tested cable proximal to the moving end undergoes successive outward and inward motions, reaching its bending limit in a single cycle, while the cable near the fixed end experiences only an outward motion. This scenario mirrors the prevalent failure seen in mining machine cables near the machine connection in underground engineering applications. Finite element simulation enables a cable segment of one pitch length to undergo bending from straight to its limit. In the reliability analysis, the numerical outcomes are compared with cables close to the heavily loaded moving end. Consequently, the results for a single bending cycle are halved, and the estimated lifespan is adjusted to $15.12e^4$ cycles, reflecting a 7% deviation from the bending test machine's bending cycles. Such consistency affirms the validity of finite element analysis in accurately predicting fatigue life.

An anatomical analysis of the shearer cable was conducted following the bending test, as depicted in Fig 27. The cable maintains an overall primary cabling twisted structure, and at $11e^4$ bending cycles, a creeping trend is observed in the red phase power conductor. This phenomenon is further substantiated through finite element simulation. Fortunately, it does not

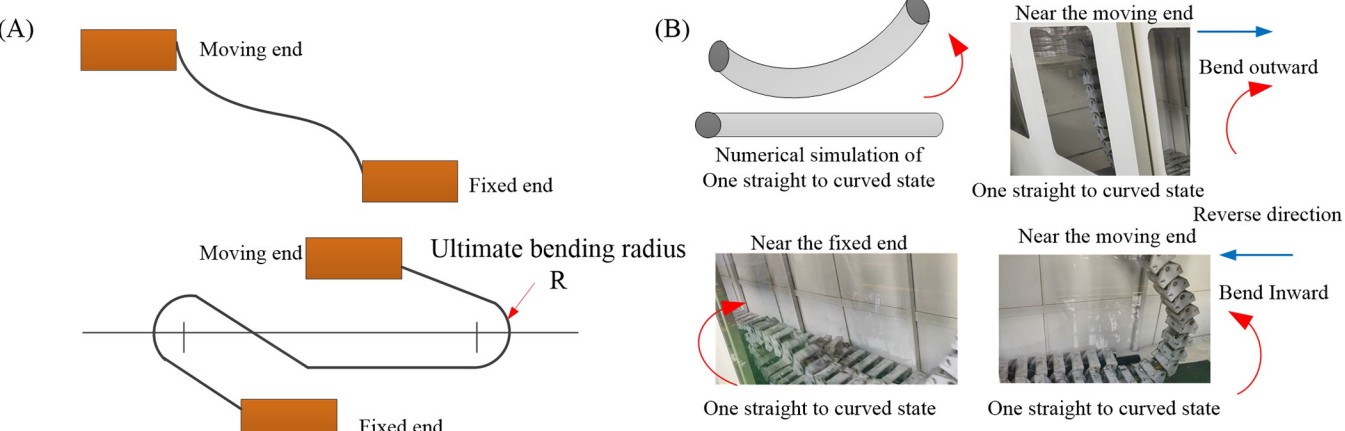

**Fig 26.** Comparison of numerical simulation and bending test on the motion state of coal mining machine cables: (A) Schematic diagram of bending test in coal industry standards; (B) Comparison of motion states.

lead to short circuits or open circuits, owing to the sliding property of the polyester tape outside the power insulation. In the examination of insulation and sheath, minimal wear is noted on the inner walls of both the sheath and insulation. During the extrusion of the control conductor, no polyester tape was applied, and the adhesive effectively immersed, achieving robust adhesion to the conductor surface. In the conductor analysis, maximum wear is concentrated at the bending areas, characterized by a substantial amount of black powder detaching from the tinned layer upon insulation removal. The polyester tape on the inner wall of the insulation experiences fractures. Additionally, there is evidence of monofilament breakage in the conductor, with the quantity not exceeding 5 strands. Bending tests on shearer cables with optimal

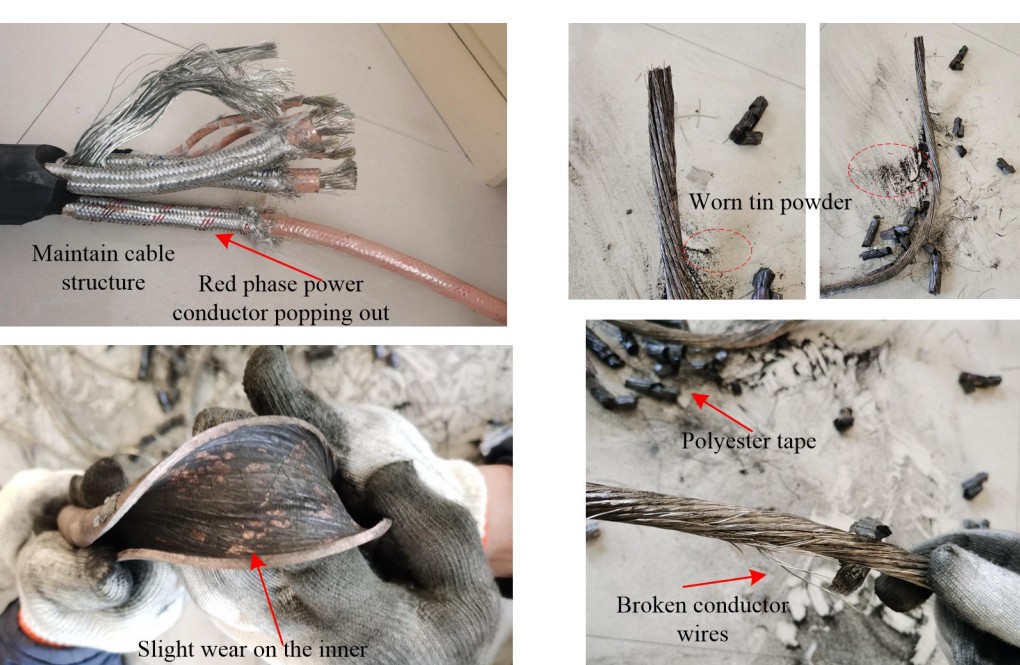

**Fig 27. Anatomy of shearer cable after bending test.**

pitch diameter ratios at different stranding levels affirm the noteworthy improvement in the mechanical performance and fatigue life of the new cable.

## 4.3 Impact of stranding direction on the mechanical properties of shearer cables

The stranding direction is categorized into left stranding and right stranding. Using a 3-layer structure as an illustration, when the stranding direction remains consistent across all layers, it is termed as parallel stranding, denoted by LLL. Conversely, if the stranding direction varies between adjacent layers, it is termed as counter stranding, denoted by LRL. A study of the influence of stranding direction on the mechanical performance of the shearer cable is undertaken across scales from monofilaments to strands and from strands to conductors, in terms of tensile and bending perspectives.

In the bending simulation, the stress distribution of power conductors with varying stranding directions is depicted in Fig 28. Counter stranding exhibits higher stress at 140.29 MPa, displaying an uneven distribution with stress concentration between layers. In contrast, parallel stranding exhibits a relatively lower stress value of 129 MPa. The consistent stranding direction of adjacent layers leads to a more uniform internal stress distribution in the conductor, thereby mitigating mutual interference and resistance between conductors. Furthermore, increased deformation flexibility contributes to the enhanced overall flexibility of the shearer cable. In the realm of industrial production, the implementation of a parallel stranding structure is attainable through a singular stranding process, thereby reducing process complexity and facilitating cost and time savings. The constriction of the outer layer upon the inner layer enhances the effectiveness of parallel stranding, resulting in the acquisition of a conductor with a diminished and denser cross-sectional outer diameter.

$$2r_x = 2R_1 \sin\left(\frac{\pi}{n}\right) \tag{19}$$

$$2r_x + g = 2R_2 \tan\left(\frac{\pi}{n+z}\right) \tag{20}$$

As depicted in Fig 29, the first layer is tangent within the layer, characterized by a monofilament radius of $r_x$ and a stranding radius of $R_1$. The inscribed regular polygon of the circle has a side length of $2r_x$. The second layer is tangent between layers, featuring a stranding radius of $R_2$. It is defined that the second layer experiences an increase of $z$ monofilaments compared to the first layer, and a gap within the second layer of the filament layer is denoted as $g$. The side lengths of the inscribed regular polygon for the two layers of the circle are expressed by Eqs (19) and (20). As the number of layers increases, maintaining a consistent layer-to-layer

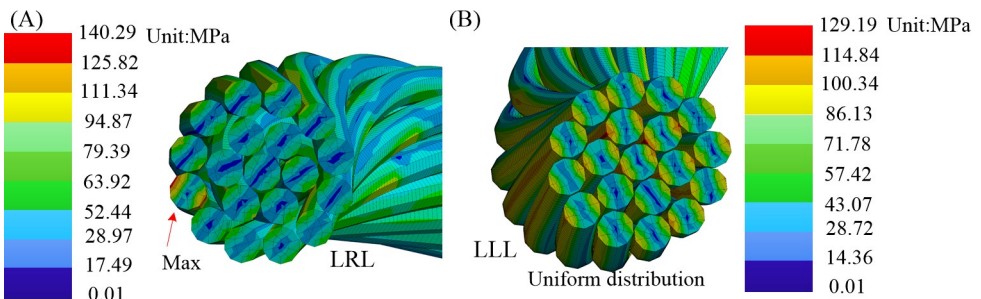

**Fig 28.** Stress of power conductors under bending conditions: (A) counter stranding LRL; (B) parallel stranding LLL.

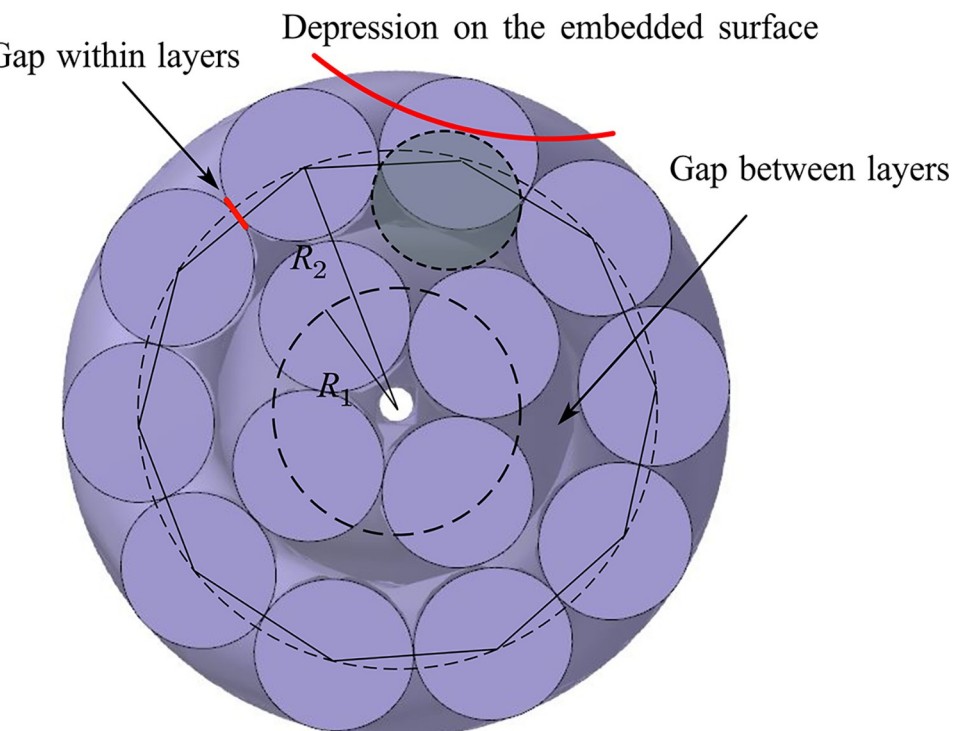

**Fig 29. Schematic diagram of multi-layers stranding structure.**

increment, the stranding radius expands, leading to a proportional growth in the gap within the outer layer monofilaments. According to the limit concept, when $n$ is sufficiently large, the circumference of a regular $n$-sided polygon can closely approximate the circumference of a circle. The arc length of the stranding pitch circle experiences an increase of $2\pi(2r_x)$, equivalent to 6.28 times the diameter of a monofilament. In consideration of the cable structure's uniformity, it is essential to ensure that the incremental rise in the number of conductor strands for each layer remains a relatively small integer. Opting for an increment of 6 is regarded as a judicious choice.

As depicted in Fig 29, a gap equivalent to 0.28 times the diameter exists in each layer. When adjacent layers share the same stranding direction, and the pitch is close or equal, the monofilaments of both the inner and outer layers assume an approximately parallel orientation. This alignment may result in the outer layer monofilaments penetrating into the gaps of the inner layer, creating depressions on the conductor surface and leading to diminished circularity. Consequently, in the context of a parallel stranding structure, it is imperative to carefully select an appropriate pitch diameter ratio to prevent the pitch sizes of adjacent layers from becoming overly close.

At the scale from monofilament to strand, the tensile strength and fracture elongation of monofilament serve as criteria for failure. Based on previous tensile tests of tinned copper strands, strains within the elastic stage remain under 1%, and the elongation at fracture is less than 30%. Longitudinal 100mm strands of LLL and LRL are tested. The low tensile rate is set at 50 mm/min, with an axial displacement of 1mm, while a high-speed tensile rate is applied at the standard upper limit of 300 mm/min, with an axial displacement of 30 mm, ensuring complete strand fracture. The force reaction derived from finite element results is related to the effective conductor cross-sectional area. When cross-sectional areas are equal, the difference

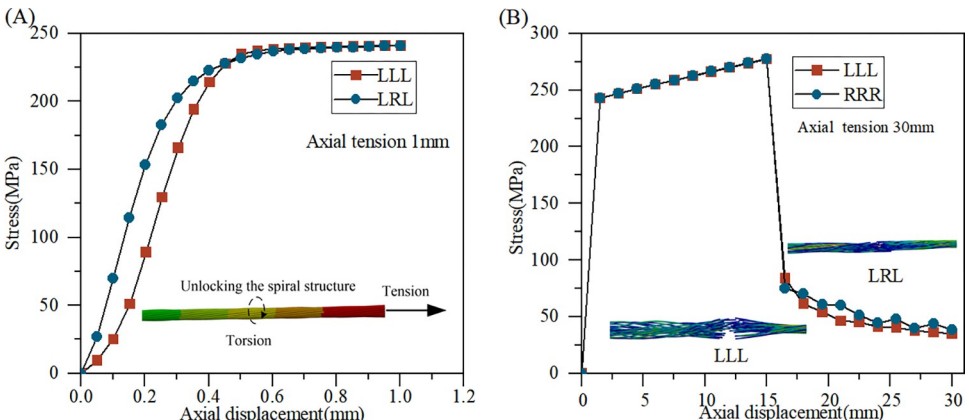

**Fig 30.** Axial tension of strands with different stranding directions: (A) tension 1 mm; (B) tension 30 mm.

in stress levels between the two types of strands during tensile simulation is insignificant compared to the bending simulation of the entire cable. However, as shown in Fig 30A, throughout the elastic stage, the stranding structure induces torsional deformation in the strand while subjected to axial tension, releasing stress through large deformation, the stress level in parallel stranding is lower than that in counter stranding. The stress curve exhibits a smaller slope, suggesting that the equivalent elastic modulus of parallel stranding is smaller. In the yield stage, the behaviors of the two types of strands closely align. In Fig 30B, the 1mm displacement during the elastic stage is inadequate for 30mm, thus high-speed tensile testing fails to capture the elastic stage characteristics, consistent with the metal tensile test standard. Furthermore, compared to counter stranding, parallel stranding may rapidly propagate within the structure during failure or fracture, leading to the scattering of the stranding structure.

However, the manufacturing process of counter stranding stranding is more complex and can only be achieved through layer-by-layer stranding. Despite experiencing slightly higher stress, the overall structure of the cable is more stable, preventing the occurrence of depressions beneath the outer layer monofilaments and ensuring improved sectional roundness. The relatively independent stranding structures of each layer exert a specific inhibitory effect on diffusion, reducing the likelihood of a failure or fracture in a particular layer propagating throughout the entire structure.

## 4.4 Impact of monofilaments on the mechanical properties of shearer cable

To maintain a consistent effective cross-sectional area of the conductor, various diameters of monofilaments are chosen to fill the cross-section. Similarly, at the scales from monofilament to strand and from strand to conductor, the impact of monofilament unit on the mechanical performance of the shearer cable is investigated in terms of tensile and bending perspectives.

At the scale from monofilaments to strands, strands and conductors are formed by using monofilaments with outer diameters of 0.39 mm and 0.5 mm. The original structure of power conductor is 24×0.5 mm(2+8+14). While ensuring a constant effective cross-sectional area of the strand conductor, a new monofilament of 0.39 mm is chosen. Following the law of inter-layer increments being 6, the new structure is determined as 40×0.39 mm(1+7+13+19). The ratio of effective conductor area to geometric area, defined as the cross-sectional filling rate, is calculated to be 74.742% for the 0.39 mm monofilament and 66.671% for the 0.5 mm monofilament. This indicates that the finer the monofilament, the higher the filling rate of the conductor cross-section.

As illustrated in Fig 31, under axial tension of 1 mm, the tensile strength of both types of strands is equivalent, with the stress distribution across the cross-section being more pronounced at the central layers and decreasing towards the outer layers. During the elastic stage, strands composed of finer monofilaments and higher filling rates demonstrate reduced stress levels, resulting in a more uniform stress distribution. Under the condition of an equivalent effective conductor cross-sectional area, higher filling rates lead to diminished clearances. Smaller diameters of monofilaments correspond to a greater number of standing layers, enhancing the flexibility of the strand. When exposed to external forces, a relatively compact structure minimizes the relative displacement between monofilaments, thereby mitigating stress concentration within the strands.

At the scale from strands to the conductors, the original structure of power strand is 24×0.5mm, and the conductor is 24 (1+6+12), with a cross-sectional filling rate of 75.759%. The new power strand is 48×0.5mm, and the structure of conductor is 10 (2+8), with a cross-sectional filling rate of 62.5%. As shown in Fig 32, the cable bending simulation results reveal

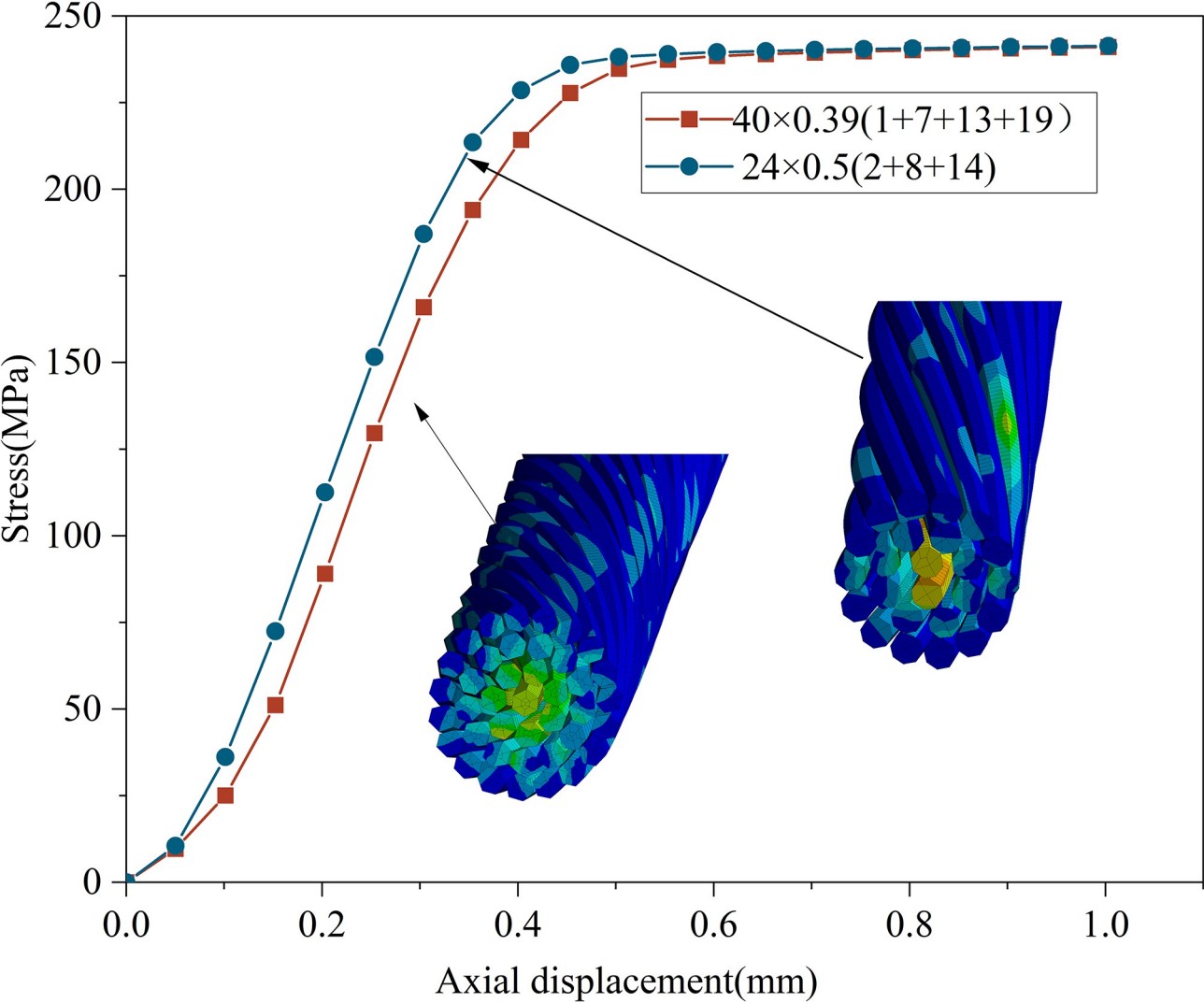

**Fig 31. Axial tension of strands composed of different monofilaments.**

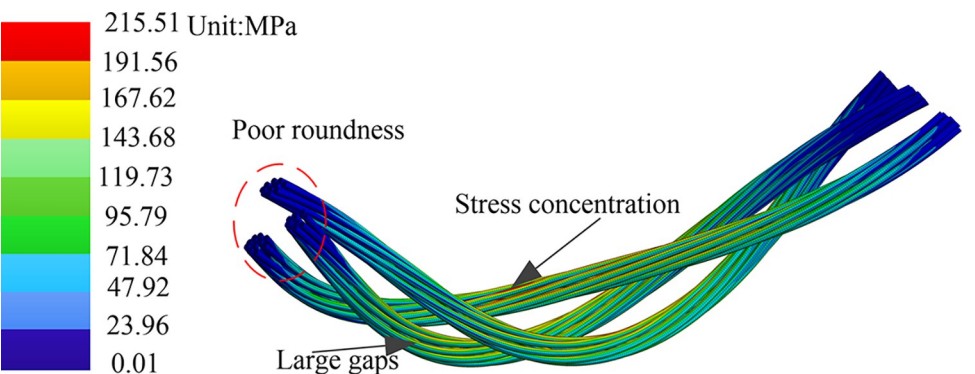

**Fig 32. Bending stress with the conductors structure of 10(2+8) and strands structure of 48×0.5 mm.**

that compared to the original power conductor which had a stress level of 183.04 MPa, the use of a thicker strand structure leads to a significantly increased maximum equivalent stress of 215.51 MPa. During the bending process, the larger gap ratio between the thicker strands causes poor circularity of the cross-section and may lead to stress concentration.

## 5. Conclusion

Utilizing parametric modeling and finite element methods, from the perspectives of tensile and bending properties, this study explores the influence of varying pitch diameter ratios at different stranding levels, stranding directions, and monofilament units on mechanical performance of shearer cables at the scale ranging from monofilaments to strands and from strands to conductors. The following conclusions can be drawn:

1. Based on the advantages of Rhino Grasshopper, this paper addresses several issues in the parametric modeling of shearer cables, including determining tangency within and between conductor layers, recursively generating spiral curves from the $(n\text{-}1)$-th level to the $n$-th level, and constructing irregular surfaces for insulation and sheathing. Rapidly modifying parameters such as pitch diameter ratios, stranding directions, and monofilament units at different stranding levels enables the efficient generation of geometrically accurate cable models adaptable to finite element simulation. This approach holds significant engineering application value for optimizing cable structures.

2. By constraining the ultimate angles of the cable clamps, the shearer cable experiences the ultimate state of bending when it reaches its minimum bending radius. The research investigated the impact of pitch diameter ratios at different stranding levels on the mechanical performance of shearer cables. The results indicate that the stress of the power and control conductors increases nonlinearly with the increase of pitch diameter ratio, and the impact on stress varies at different stranding levels. The optimal combination was determined: a pitch diameter ratio of 6 for cabling, 5 for the control conductor, and 14 for the power conductor. The stress level of the power conductor is 183 MPa, while that of the control conductor is 120 MPa. The fatigue life of the improved cable predicted by Ncode aligns with the results of the bending test, demonstrating a capability of up to $15.12e^{4}$ cycles. This also validates influence of structural parameter changes on the mechanical properties of cables can be evaluated from perspectives of finite element stress level analysis and fatigue life prediction.

3. Under tensile and bending conditions, the stress in the parallel stranding is smaller and evenly distributed. The similar pitch tends to make upper-layer monofilaments embedded in the gap of the inner layer, and the fracture propagates rapidly and is prone to the scattering of the stranding structure. It is recommended to carefully choose an appropriate pitch diameter ratio to prevent close proximity of pitches in adjacent layers. The production process of counter stranding is complex, and although it experiences higher stress, it ensures better roundness and a more stable structure. Regarding the selection of monofilaments, when maintaining a constant effective conductor cross-sectional area, the finer the composition of monofilament units, the higher the cross-sectional fill rates. A relatively compact structure can reduce relative displacement, decrease stress concentration within the strands, and achieve improved tensile and bending performance.

## Acknowledgments

The authors would like to acknowledge the support and contribution from the State Key Lab of Mining Machinery Engineering of Coal Industry, Liaoning Technical University, China. The authors would also like to thank Changlong cable factory of Shandong energy group for providing cable parameters and experimental conditions.

## Author Contributions

**Conceptualization:** Lijuan Zhao.

**Data curation:** Haining Zhang, Shijie Yang.

**Formal analysis:** Haining Zhang.

**Funding acquisition:** Lijuan Zhao.

**Investigation:** Feng Gao.

**Methodology:** Lijuan Zhao, Haining Zhang.

**Resources:** Feng Gao.

**Software:** Haining Zhang.

**Supervision:** Feng Gao, Shijie Yang.

**Validation:** Lijuan Zhao, Haining Zhang, Shijie Yang.

**Writing – original draft:** Lijuan Zhao, Haining Zhang.

**Writing – review & editing:** Lijuan Zhao, Haining Zhang.

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
