## [Decision Letter · Decision Letter 0]

16 Apr 2024

PONE-D-24-10993Research on p arameterized modeling and mechanical characteristics  of shearer  cablesPLOS ONE

Dear Dr. Zhang,

Thank you for submitting your manuscript to PLOS ONE. After careful consideration, we feel that it has merit but does not fully meet PLOS ONE’s publication criteria as it currently stands. Therefore, we invite you to submit a revised version of the manuscript that addresses the points raised during the review process.

We look forward to receiving your revised manuscript.

Kind regards,

Khalil Abdelrazek Khalil, Ph.D.

Academic Editor

PLOS ONE

Reviewers' comments:

Reviewer's Responses to Questions

**Comments to the Author**

1. Is the manuscript technically sound, and do the data support the conclusions?

Reviewer #1: Yes

Reviewer #2: Yes

2. Has the statistical analysis been performed appropriately and rigorously? 

Reviewer #1: No

Reviewer #2: Yes

3. Have the authors made all data underlying the findings in their manuscript fully available?

Reviewer #1: Yes

Reviewer #2: Yes

4. Is the manuscript presented in an intelligible fashion and written in standard English?

Reviewer #1: Yes

Reviewer #2: Yes

5. Review Comments to the Author

Reviewer #1: The study focuses on predicting the mechanical performance and fatigue life of shearer cables under harsh conditions. It addresses the challenges of parametric modeling and examines variations in pitch diameter ratios, stranding levels and monofilaments. Although this article provides useful information for modeling the structure of shearer cables, the following comments must be considered for the article to be accepted:

1. The parameters of the model shown in schematic Figures 2 and 3 are not sufficiently clear and have not been well defined and represented.

2. Considering existing models for shearer cable modeling, the innovation of the modeling presented in this article is questionable. Authors should transparently discuss the advantages and capabilities of Rhino Grasshopper modeling compared to previous works.

3. 3. Statistically, reliable determination of mechanical properties requires repeatability of tests under consistent conditions. Regarding Figure 11, explanations need to be added to the text regarding 4 repetitions of tensile tests for control KA6, 3 repetitions for control KA10 and 5 repetitions for Power 24*0.5 and Power 48*0.5. Presentation of the average experimental data along with the standard deviation in Tables 1 and 2 is also important. The standards, loading rates and load applications used in the tensile tests are not specified.

4. The characteristics of the bilinear isotropic hardening model are not provided in the numerical simulations!

5. The observed decreasing trend in stress values for Power Strand, Control Strand, and Ground Strand depicted in Figures 14 and 17 requires thorough interpretation and analysis to understand its significance in the context of the study.

6. The R-Squared value for the fitted curves in figure 18 should be reported.

7. Section 4-2 does not validate the fatigue life prediction results! How can we trust simulation results without validation? Further details on flexural fatigue loading should be provided. What was the mean stress and the stress amplitude? How was Ncode's predicted fatigue life corrected to 15.12e4 cycles and how was the 7% error determined?

8. In Figure 27, what were the tensile loading rates for 1 mm and 30 mm displacement? Why is the stress level in parallel stranding in the elastic stage at a low axial tension of 1 millimeter lower than in counter stranding? Why are the two types of twisting identical at a higher axial tension of 30 mm?

9. In summary, there is a lack of research reports on the optimized modeling parameters required to achieve the desired mechanical properties in shearer cables. This gap affects the completeness and applicability of the study's results in the field of shearer cable modeling and design.

Reviewer #2: The manuscript is generally well written.

Abstrcut, Introduction, 2 and 3 are well written.

1-However, while interpreting the results in chapter 4, it would be better if the differences and similarities were emphasized by giving examples from studies conducted in the literature. Thus, the number of literature will also increase.

2-Also, it would be better if some numerical results obtained were given in the conclusion section.

If these corrections are made, it is appropriate to publish.

6. PLOS authors have the option to publish the peer review history of their article (what does this mean?). If published, this will include your full peer review and any attached files.

Reviewer #1: No

Reviewer #2: No

---

## [Author Response · Author response to Decision Letter 0]

25 Apr 2024

Dear Editor and Reviewers:

On behalf of my co-authors, we are grateful to you for giving us an opportunity to revise our manuscript. We appreciate you very much for your positive and constructive comments and suggestions on our manuscript entitled “Research on parameterized modeling and mechanical characteristics of shearer cables”([PONE-D-24-10993] - [EMID:0e19621c9ab36903]).

We have studied reviewers’ comments carefully and tried our best to revise our manuscript according to the comments. The following are the responses and revisions I have made in response to the reviews’ questions and suggestions on an item-by-item basis. Thanks again to the hard work of editor and reviewer!

Response to the comments of Editor:

Comment No.1：Please ensure that your manuscript meets PLOS ONE's style requirements, including those for file naming. 

Response: Based on the requirements of the journal formatting requirements document you shared, we have revised the format of the manuscript to ensure that it meets the requirements for journal publication, including headings, references, etc.

Comment No.2：Please note that PLOS ONE has specific guidelines on code sharing for submissions in which author-generated code underpins the findings in the manuscript. In these cases, all author-generated code must be made available without restrictions upon publication of the work. Please review our guidelines at https://journals.plos.org/plosone/s/materials-and-software-sharing#loc-sharing-code and ensure that your code is shared in a way that follows best practice and facilitates reproducibility and reuse.

Response: Our manuscript does not have any issues regarding code sharing. The data from the tensile tests has been uploaded to a public database at the time of our initial manuscript submission, openly available for scholars to access and utilize for learning purposes.

Comment No.3：Please review your reference list to ensure that it is complete and correct. If you have cited papers that have been retracted, please include the rationale for doing so in the manuscript text, or remove these references and replace them with relevant current references. Any changes to the reference list should be mentioned in the rebuttal letter that accompanies your revised manuscript. If you need to cite a retracted article, indicate the article’s retracted status in the References list and also include a citation and full reference for the retraction notice.

Response: We have supplemented and updated the references according to the journal requirements, and correctly numbered them in the text, without any retracted papers in the references.

Comment No.4：While revising your submission, please upload your figure files to the Preflight Analysis and Conversion Engine (PACE) digital diagnostic tool, https://pacev2.apexcovantage.com/. PACE helps ensure that figures meet PLOS requirements. To use PACE, you must first register as a user. Registration is free. Then, login and navigate to the UPLOAD tab, where you will find detailed instructions on how to use the tool. If you encounter any issues or have any questions when using PACE, please email PLOS at figures@plos.org. Please note that Supporting Information files do not need this step.

Response: According to the journal's requirements, we have uploaded all the images from the manuscript to the PACE website, corrected them to meet the journal's requirements for TIFF format images, renamed them as required, and replaced them in the revised manuscript submitted again. These images have been packaged into a compressed file attached to the submission system, with the filename "figures corrected by PACE"

Response to the comments of Reviewer #1

Comment No.1: The parameters of the model shown in schematic Figures 2 and 3 are not sufficiently clear and have not been well defined and represented.

Response:According to your suggestion, we have redrawn Figures 2 and 3, providing more detailed and accurate explanations of the diagrams and definitions of variables. Additionally, we have corrected the error in the title of Figure 3. The relevant modifications and additions are marked in red in the main text, specifically on pages 4 and 5.

Comment No.2:Considering existing models for shearer cable modeling, the innovation of the modeling presented in this article is questionable. Authors should transparently discuss the advantages and capabilities of Rhino Grasshopper modeling compared to previous works.

Respone: Following your suggestion, we have supplemented the introduction with references to three studies on the shearer cables, highlighting their merits and limitations, and transparently discussing the advantages and capabilities of Rhino Grasshopper modeling compared to previous works. 

Figure 2 was added, illustrates the parameterized modeling of shearer cables based on Rhino Grasshoppe，including displays 3D models on the Rhino interface and visual programming in the Grasshopper interface. 

In contrast to previous research on shearer cable modeling, this paper introduces a parameterized modeling approach utilizing Rhino Grasshopper, which enhances modeling accuracy and efficiency. This method retains more geometric structural intricacies of the cables compared to oversimplified models. It employs visual programming to construct recursion from n-1 to n levels, eliminating the need for intricate high-order spiral curve calculations and assembly, thus facilitating convenient parameter adjustments. Furthermore, it conducts quantitative assessments of rubber sheath and insulation deformation during extrusion and cable formation, enabling precise establishment of irregular curved surface structures. These revisions and additions have been marked in red in the main text, specifically on pages 2 and 4.

Comment No.3: Statistically, reliable determination of mechanical properties requires repeatability of tests under consistent conditions. Regarding Figure 11, explanations need to be added to the text regarding 4 repetitions of tensile tests for control KA6, 3 repetitions for control KA10 and 5 repetitions for Power 24*0.5 and Power 48*0.5. Presentation of the average experimental data along with the standard deviation in Tables 1 and 2 is also important. The standards, loading rates and load applications used in the tensile tests are not specified.

Respone：According to your suggestion, We have made corrections to improve the accuracy and standardization of our tensile testing procedure.

According to the national standard GB/T 2951.11-2008, the movement speed of the clamp in the tensile test of rubber insulation and sheath is (250±50) mm/min, and the tensile speed of YN21003 was set to 200 mm/min.

When studying copper strands, in compliance with national standards GB/T 4909.3-2009 and GB/T 228-2002, the tensile speed of the AI-7000-LA20 tensile testing machine was set to 50 mm/min.

Standardized the uniformity of the sample numbers in the experimental groups and supplemented the average and standard deviation of the data. Removed irrelevant information about the KA10 control strands from this paper, retaining the fifth set of test data for the KA6 control core wire from the original dataset, with a sample size of 5 in each experimental group.

The abbreviations AVG and SD in Table 1and 2 represent the mean and standard deviation.

The above supplements and modifications have been marked in red in the main text, specifically on pages 8 and 9.

Comment No.4： The characteristics of the bilinear isotropic hardening model are not provided in the numerical simulations!

Respone: According to your suggestion, we have added Figure 13, which depicts the Schematic Diagram of the Bilinear Isotropic Hardening model, providing detailed explanations. The characteristics of the bilinear isotropic hardening model were not provided in the numerical simulations.

The material's elastic-plastic deformation is simulated using a bilinear isotropic hardening model, which incorporates two linear segments. One segment represents the elastic behavior with a slope equivalent to the elastic modulus, while the other denotes the plastic behavior with a slope equivalent to the tangent modulus. This approach effectively captures the material's nonlinear characteristics while improving simulation efficiency. Implementation of the Bilinear Isotropic Hardening model within finite element software material libraries requires specifying parameters such as the elastic modulus, yield strength, tangent modulus, and tensile strength.

The above supplements and modifications have been marked in red in the main text, specifically on pages 9.

Comment No.5：The observed decreasing trend in stress values for Power Strand, Control Strand, and Ground Strand depicted in Figures 14 and 17 requires thorough interpretation and analysis to understand its significance in the context of the study.

Respone: We have drawn Figure 19, which, combined with Figure 16, elucidates the stress distribution along the arc length of the cable core from the perspectives of loading conditions, motion processes, and classical models. It demonstrates a pattern of higher stress in the middle and lower stress at both ends, consistent with theoretical mechanics analysis. The simulation results align with theoretical mechanics analysis and explain the reasons for the fluctuations observed.

Utilizing the loading and displacement method depicted in Fig. 16, successive cable clamps achieved a 30-degree incremental rotation, culminating in the cable reaching the ultimate bending state depicted in Fig. 19A. This study examines the stress distribution along the axial arc length of the cable, employing the ideal simply supported beam model illustrated in Fig. 19B. When the cable is subjected to the ultimate bending state, the inner wall of the cable clamp applies a uniformly distributed load q along the axial length l of the cable. Applying the equilibrium equation, shear forces are opposite at both ends, reaching a maximum of ql/2. At the midpoint, the shear force diminishes to zero, while the bending moment reaches its maximum value at the midpoint, ql2/8, with both ends registering zero. Additionally, since stress magnitude is directly proportional to the bending moment, stress distribution in the ultimate bending state of cables adheres to the pattern of being greater in the middle and smaller at both ends along the arc length.

In comparison to the ideal mechanical model, the simulated stranded wire experiences more complex combined loads in the simulation, leading to fluctuations in the curves. Nonetheless, it also demonstrates the distribution pattern of higher stress levels in the middle and lower stress levels at both ends.

The above supplements and modifications have been marked in red in the main text, specifically on pages 13and 14.

Comment No.6：The R-Squared value for the fitted curves in figure 18 should be reported.

Respone: We have retrieved the coefficient of determination R-Squared value from the original fitting files of four curves in Origin software and supplemented it at the corresponding locations in the article, 0.997 for y1, 0.989 for y3, 0.995 for y2, 0.994 for y4, 0.907 for y5, 0.992 for y6.

These additions have been marked in red on page 15.

Comment No.7：Section 4-2 does not validate the fatigue life prediction results! How can we trust simulation results without validation? Further details on flexural fatigue loading should be provided. What was the mean stress and the stress amplitude? How was Ncode's predicted fatigue life corrected to 15.12e4 cycles and how was the 7% error determined?

Respone: Our numerical simulation's lifetime prediction results have been compared with those from the shearer cable bending test machine, validating their reasonableness. 

We have provided details of fatigue life analysis, including the determination of parameters such as stress range amplitude and the first fatigue strength exponent. Additionally, we highlighted the advantages and rationality of using the finite element results' "step time" as a load spectrum. Figure 26 has been generated to analyze the corrections of the numerical simulation from the perspectives of loading conditions and motion processes.

Utilizing finite element analysis results and time-step load spectra automatically generated by the software, stress history is derived without the necessity of calculating average stress or stress amplitude for load spectrum generation. This approach is applicable for transient and large deformation explicit finite element solution results.

Figure 26A illustrates that in accordance with the coal industry standard MT 818.1-2009, the bending test machine subjects the tested cable section to a bending process transitioning from a straight to an S-shaped state during each reciprocating cycle. Conversely, as depicted in Figure 26B, the tested cable proximal to the moving end undergoes successive outward and inward motions, reaching its bending limit in a single cycle, while the cable near the fixed end experiences only an outward motion. This scenario mirrors the prevalent failure seen in mining machine cables near the machine connection in underground engineering applications. Finite element simulation enables a cable segment of one pitch length to undergo bending from straight to its limit. In the reliability analysis, the numerical outcomes are compared with cables close to the heavily loaded moving end. Consequently, the results for a single bending cycle are halved, and the estimated lifespan is adjusted to 15.12e4 cycles, reflecting a 7% deviation from the bending test machine's bending cycles.

The above supplements and modifications have been marked in red in the main text, specifically on pages 16and 18.

Comment No.8： In Figure 27, what were the tensile loading rates for 1 mm and 30 mm displacement? Why is the stress level in parallel stranding in the elastic stage at a low axial tension of 1 millimeter lower than in counter stranding? Why are the two types of twisting identical at a higher axial tension of 30 mm?

Respone: Based on previous tensile tests of tinned copper strands, strains within the elastic stage remain under 1%, and the elongation at fracture is less than 30%. 

Longitudinal 100mm strands of LLL and LRL are tested. The low tensile rate is set at 50 mm/min, with an axial displacement of 1mm, while a high-speed tensile rate is applied at the standard upper limit of 300 mm/min, with an axial displacement of 30 mm, ensuring complete strand fracture. 

When cross-sectional areas are equal, the difference in stress levels between the two types of strands during tensile simulation is insignificant compared to the bending simulation of the entire cable.However, as shown in Fig 30A, throughout the elastic stage, the stranding structure induces torsional deformation in the strand while subjected to axial tension, releasing stress through large deformation, the stress level in parallel stranding is lower than that in counter stranding. The stress curve exhibits a smaller slope, suggesting that the equivalent elastic modulus of parallel stranding is smaller. In the yield stage, the behaviors of the two types of strands closely align.

In Fig 30B, the 1mm displacement during the elastic stage is inadequate for 30mm, thus high-speed tensile testing fails to capture the elastic stage characteristics, consistent with the metal tensile test standard.

The above supplements and modifications have been marked in red in the main text, specifically on page21.

Comment No.9：In summary, there is a lack of research reports on the optimized modeling parameters required to achieve the desired mechanical properties in shearer cables. This gap affects the completeness and applicability of the study's results in the field of shearer cable modeling and design.

Respone: According to your suggestion, at the end of chapter 4.1, we have added a comparison with previous studies to demonstrate the progressiveness of our research.

Previous studies have explored the impact of primary cable stranding diameter ratios on cable tensile and bending performance, overlooking the holistic influence of secondary dynamic and control cable strand diameter ratios. The bending tests typically employed three-point bending or direct cable model b

---

## [Decision Letter · Decision Letter 1]

6 May 2024

Research on parameterized modeling and mechanical characteristics of shearer cables

PONE-D-24-10993R1

Dear Dr. Zhang,

We’re pleased to inform you that your manuscript has been judged scientifically suitable for publication and will be formally accepted for publication once it meets all outstanding technical requirements.

Kind regards,

Khalil Abdelrazek Khalil, Ph.D.

Academic Editor

PLOS ONE

Additional Editor Comments (optional):

Reviewers' comments:

Reviewer's Responses to Questions

**Comments to the Author**

1. If the authors have adequately addressed your comments raised in a previous round of review and you feel that this manuscript is now acceptable for publication, you may indicate that here to bypass the “Comments to the Author” section, enter your conflict of interest statement in the “Confidential to Editor” section, and submit your "Accept" recommendation.

Reviewer #1: All comments have been addressed

Reviewer #2: All comments have been addressed

2. Is the manuscript technically sound, and do the data support the conclusions?

Reviewer #1: Yes

Reviewer #2: Yes

3. Has the statistical analysis been performed appropriately and rigorously? 

Reviewer #1: Yes

Reviewer #2: Yes

4. Have the authors made all data underlying the findings in their manuscript fully available?

Reviewer #1: Yes

Reviewer #2: Yes

5. Is the manuscript presented in an intelligible fashion and written in standard English?

Reviewer #1: Yes

Reviewer #2: Yes

6. Review Comments to the Author

Reviewer #1: The authors have satisfactorily addressed all my concerns in my previous reviews. Hence, the manuscript is recommended for publication in PLOS ONE in its current form.

Reviewer #2: It is a good study in terms of experimental work and numerical study. Requested corrections have been made, the manuscript can be published.Requested corrections have been made, the manuscript can be published.

7. PLOS authors have the option to publish the peer review history of their article (what does this mean?). If published, this will include your full peer review and any attached files.

Reviewer #1: No

Reviewer #2: No

---

## [Editor Report · Acceptance letter]

10 May 2024

PONE-D-24-10993R1 

PLOS ONE

Dear Dr. Zhang, 

I'm pleased to inform you that your manuscript has been deemed suitable for publication in PLOS ONE. Congratulations! Your manuscript is now being handed over to our production team.

Kind regards, 

on behalf of

Dr. Khalil Abdelrazek Khalil 

Academic Editor

PLOS ONE